# Vitamin D, Th17 Lymphocytes, and Breast Cancer

**DOI:** 10.3390/cancers14153649

**Published:** 2022-07-27

**Authors:** Beata Filip-Psurska, Honorata Zachary, Aleksandra Strzykalska, Joanna Wietrzyk

**Affiliations:** Hirszfeld Institute of Immunology and Experimental Therapy, Polish Academy of Sciences, Rudolfa Weigla St. 12, 53-114 Wroclaw, Poland; beata.filip-psurska@hirszfeld.pl (B.F.-P.); honorata.zachary@hirszfeld.pl (H.Z.); aleksandra.strzykalska@hirszfeld.pl (A.S.)

**Keywords:** vitamin D, calcitriol, Th17 lymphocytes, dendritic cells, breast cancer

## Abstract

**Simple Summary:**

The effect of vitamin D_3_ on the development of breast cancer (favorable, ineffective, or even unfavorable) depends on many factors, such as age, menopausal status, or obesity. The immunomodulatory effect of vitamin D may be unfavorable in case of breast cancer progression. The effect of vitamin D on Th17 cells may depend on disease type and patients’ age. Our goal was to summarize the data available and to find indications of vitamin D treatment failure or success. Therefore, in this review, we present data describing the effects of vitamin D_3_ on Th17 cells, mainly in breast cancer.

**Abstract:**

Vitamin D_3_, which is well known to maintain calcium homeostasis, plays an important role in various cellular processes. It regulates the proliferation and differentiation of several normal cells, including immune and neoplastic cells, influences the cell cycle, and stimulates cell maturation and apoptosis through a mechanism dependent on the vitamin D receptor. The involvement of vitamin D_3_ in breast cancer development has been observed in numerous clinical studies. However, not all studies support the protective effect of vitamin D_3_ against the development of this condition. Furthermore, animal studies have revealed that calcitriol or its analogs may stimulate tumor growth or metastasis in some breast cancer models. It has been postulated that the effect of vitamin D_3_ on T helper (Th) 17 lymphocytes is one of the mechanisms promoting metastasis in these murine models. Herein we present a literature review on the existing data according to the interplay between vitamin D, Th17 cell and breast cancer. We also discuss the effects of this vitamin on Th17 lymphocytes in various disease entities known to date, due to the scarcity of scientific data on Th17 lymphocytes and breast cancer. The presented data indicate that the effect of vitamin D_3_ on breast cancer development depends on many factors, such as age, menopausal status, or obesity. According to that, more extensive clinical trials and studies are needed to assess the importance of vitamin D in breast cancer, especially when no correlations seem to be obvious.

## 1. Introduction

Despite great advances in science and medicine over the last few decades, breast cancer remains the most frequently diagnosed cancer among women around the world. The factors favoring breast cancer development include hormonal disorders, early menstruation, an abnormal number of menstrual cycles during life, pregnancy, obesity, diet, hormonal replacement therapy, lack of physical activity, and lifestyle or genetic disorders (the latter accounting for up to 10% of all cases) [1,2]. Throughout her life, the body of a woman constantly experiences changes in the concentration of hormones, which regulate many processes in a single cell as well as in tissues. Due to irregular menstrual cycles, early menstruation, late motherhood or absence of pregnancies, or the supply of exogenous estrogens via food or hormonal replacement therapy, tissues are exposed to excessive levels of estrogens, which results in cell cycle dysregulation and excessive cell proliferation, leading to the development of cancer. On the other hand, after menopause, when estrogen synthesis by ovaries stops and the process is taken over by other tissues due to the aromatase found in their cells, this enzyme involved in estrogen production may be excessively activated by external factors, promoting the development of breast cancer [3,4].

The development of breast cancer, as with all cancers, is a complex and long-term process. The cell consolidates successive genetic changes in each stage, and eventually transforms from an originally healthy, normal state into a malignant, invasive neoplastic one [4]. Breast neoplasms are very diverse. During each stage of disease development, the expression level of various proteins changes, which is often related to the variable sensitivity of these tumors to the applied treatment. Breast cancers are classified, inter alia, based on the expression of receptors such as estrogen receptor α (ERα; often referred to as ER), progesterone receptor (PR), and human epidermal growth factor receptor (EGFR; also known as ErbB2/HER2) with tyrosine kinase activity [3,4,5,6]. The beta form of the estrogen receptor (ERβ) is also known; however, only the expression of ERα is considered in the clinical classification of cancer due to the understanding of the function of this receptor and its significance in the development of breast tumors. The role of ERβ in breast cancer development is not fully understood [6,7], although researchers have been showing increasing interest in studying its functions in recent years [8,9,10,11,12] because of its potential of novel therapies in ERα negative-BCa and TNBC [5]. Reports also indicate the use of ERα-36, an isoform of ERα, as a diagnostic marker in breast cancer. The presence of this receptor in neoplastic cells, apart from the “proper” ERα (Erα-66), causes disturbances in signal transmission. ERα-36, similarly to GPR-30 (G protein-bound estrogen receptor), performs signal transduction bypassing the genomic pathway, while blocking signaling in the cell via ERα. In patients diagnosed with hormone-dependent cancer, ERα-36 interferes with hormonal therapy and reduces its effectiveness [13,14].

Approximately 60–70% of breast cancers may express ER and/or PR and are called luminal A subtype (ER+ and/or PR+, HER2−, Ki-67 low) and luminal B subtype (ER+ and/or PR+, HER2+ or HER2−but high Ki-67), approximately 20–30% overexpress HER2 (HER2 enriched), and approximately 10% do not express any of the three receptors and are classified as “triple negative” (basal-like tumors) [6,15]. During diagnosis of these neoplasms, the expression levels and activity of aromatase, an enzyme from the cytochrome P450 (CYP) group, which is involved in a stage of estrogen synthesis, are also assessed. In patients with ER+ cancer, aromatase is strongly overexpressed to provide cells with adequate estrogens for their interrupted growth [5,6,7]. On the other hand, in patients with triple-negative breast cancer (TNBC), aromatase present in other tissues (such as ovary) promotes metastasis of cancer cells to different organs [16].

The expression of ER, PR, and HER2 is altered in both early and later stages of the development of breast cancer. At the early stage, the neoplasm is referred to as in situ tumor, which forms proliferating neoplastic epithelial cells. Breast cancer can be classified at the in situ stage: ductal carcinoma in situ (DCIS) and lobular carcinoma in situ. In the further stages of cancer development, the tumors infiltrate the surrounding tissues and become invasive cancers, metastasizing to distant organs [2,17].

In addition to proteins that are directly involved in the regulation of endocrine metabolism and growth factor production, breast cancer cells also overexpress proteins related to vitamin D metabolism and signaling via the vitamin D receptor (VDR) or calcium-sensing receptor (CaSR). Studies on the expression of CYP27B1, CYP24A1 (vitamin D-metabolizing enzymes), and VDR in sections of mammary gland tissues taken from women suffering from breast cancer of various stages and healthy women have shown that imbalance between the expression of these CYP enzymes and decreased VDR expression promote invasiveness and metastasis of tumor [18,19]. Impaired regulation of calcium and phosphate metabolism, and disturbances in the related signal transduction pathways, leads to the development of many diseases, including abnormal changes in the mammary gland tissue and subsequently cancer [20,21,22]. However, these observations from scientific research are yet to be applied in the clinical classification of tumors.

## 2. Vitamin D and Breast Cancer Prevention and Treatment

In the 1920s, sunbathing and consumption of fish oil were prescribed to children who were at risk of developing rickets. Several observations highlight the wide range of effects of vitamin D on the human body. The influence of sunlight, latitude, and a vitamin D-rich diet on the development of many civilization diseases, including breast cancer, has been studied [23]. Numerous epidemiological studies have shown a direct relationship between geographic location and decreasing UV-B exposure as the distance from the equator increases. Low vitamin D levels due to low exposure to UV radiation, as well as consumption of a vitamin D-deficient diet, are associated with an increased risk of colon, breast, ovary, and prostate cancers [17,18,19,20,24].

A study analyzed the serum level of 25-hydroxyvitamin D (25(OH)D; the main metabolite used to assess the vitamin D status in the body) in pre- and postmenopausal women, as well as genetic predisposition, such as VDR gene polymorphism and expression of vitamin D-metabolizing enzymes at various stages of cancer development [25].

The study revealed that higher serum 25(OH)D is strongly associated with better prognosis. Moreover, it was supposed that lower vitamin D serum levels in the summertime may speak for low vitamin D levels during the whole year. This in turn is of great importance according to the crucial role of vitamin D in various cellular processes, apart from maintaining calcium homeostasis, such as the regulatory processes of proliferation and differentiation of many normal and neoplastic cells, influence on the cell cycle, and stimulation of cell maturation and apoptosis through a mechanism dependent, among others, on VDR [26,27,28]. Vitamin D controls angiogenesis, influences signaling pathways, that are involved in migration and metastasis of cancer cells to distant tissues and organs, for example, by regulating the expression of adhesion molecules on the surface of these cells [7,10,29,30,31,32,33]. Therefore, vitamin D is considered an important regulator of life processes in the human body.

### 2.1. Role of Vitamin D in Development of Breast Cancer

#### 2.1.1. The 25-Hydroxyvitamin D Level and Cancer Correlations

Some clinical studies have analyzed the role of vitamin D in the pathogenesis of breast cancer, but not all of them support the protective role of vitamin D against breast cancer development. Women diagnosed with breast cancer revealed a significantly lower 25(OH)D level than healthy women. Patients with advanced or metastatic breast cancer showed significantly lower 25(OH)D status than patients with early-stage disease [21,22,23,26,27].

On the other hand, a recent meta-analysis indicated that high serum level of 25(OH)D has a significant protective effect only in premenopausal women [31] or at the time of diagnosis. The European population-based cohort studies conducted among older adults confirmed that the risk of breast cancer increased with the concentration of 25(OH)D [32]. Recently, Kanstrup et al. showed poorer breast cancer survival among women with high levels of 25(OH)D (above 110 nmol/L) [33].

#### 2.1.2. Studies on VDR and Other Vitamin D-Related Molecules

Studies on VDR expression revealed that low VDR expression in tumor was correlated with aggressive characteristics of breast cancer [28]. What’s more, a favorable tumor characteristic included smaller size, lower grade, estrogen receptor positivity and progesterone receptor positivity, and lower expression of Ki67 (overall better prognosis). Also, the nuclear and cytoplasmic VDR expression were associated with a low risk of breast cancer mortality, with hazard ratios 0.56 (95% CI 0.34–0.91) and 0.59 (0.30–1.16), respectively [29]. Moreover, the VDR expression in circulating tumor cells (CTCs) can be suggested as a potential prognostic biomarker of breast cancer [30].

Some studies that analyzed the relationships between exposure to sunlight, intake of dietary vitamin D, serum levels of 25(OH)D and calcitriol (1,25(OH)_2_D_3_, the active metabolite of vitamin D_3_), and expression and genetic variations of VDR and other vitamin D-related molecules are summarized in Table 1.

Apart from the studies concerning the relationships between the expression and polymorphism of vitamin D-associated molecules (including VDR, CYP24A1, and CYP27B1) and breast cancer, those focusing on CaSR, which regulates the release of parathyroid hormone (PTH) and calcitonin in response to changes in blood calcium levels, seem to be of great importance [79,80,81]. Together with PTH, calcitonin, and calcitriol, CaSR contributes to maintaining the calcium homeostasis in the body. This receptor is found not only in cells involved in the regulation of calcium levels, such as thyroid C cells, kidney cells, colon epithelial cells, osteoclasts, and osteoblasts, but also in cells of the brain, pancreas, and stomach [82]. Furthermore, the presence of CaSR has been observed in neoplastic cells of the breast, prostate, and colon as well as in parathyroid neoplasms; however, the level of CaSR expression in neoplastic tissues significantly differs from that in normal tissues [83,84,85]. In normal mammary tissue, CaSR is found in epithelial cells, and its expression changes as a result of changes occurring in the breast. The level of CaSR remains lower before and during pregnancy, but during the lactation period it significantly increases due to changes in the concentration of calcium ions [85,86]. CaSR expression at the mRNA and protein level has been studied in breast cancer cells. In normal breast tissue, the secretion of PTH is inhibited by an increase in the calcium level. In prostate and breast tumors, an increase in calcium enhances the secretion of PTH-related protein (PTHrP) by influencing G protein activation, as demonstrated in the MCF-7 model, which often results in hypercalcemia [87]. The release of a high concentration of calcium from bone during breast cancer metastasis is believed to activate multiple CaSR signaling pathways that regulate the growth, proliferation, and migration of cancer cells to bone [86,88,89]. It was found that increased CaSR expression and increased PTHrP level in breast cancer cells correlate with higher predisposition of these cells to metastasize and colonize bones compared to other tissues and organs, such as the brain. Mamillapalli et al. have shown that the use of anti-PTHrP antibodies in mice bearing the MDA-MB-231 human mammary carcinoma xenograft reduced the number of bone metastases [87]. In response to an increase in calcium concentration, CaSR stimulates the activation of choline kinase and the production of phosphocholine, thereby accelerating the proliferation of cancer cells and enhancing their invasiveness and resistance [90]. Moreover, observations indicate that CaSR can regulate the proliferation of tumor cells by activating EGFR [91,92]. In MCF-7 human breast cancer cells, calcitriol was found to activate apoptosis via a nongenomic pathway involving CaSR [93,94,95].

The presence of CaSR in breast cancer cells and in bones, as well as in metastases, opens up new possibilities of applying the knowledge about this receptor to regulate the disorders caused by changes in bone structure and the course of cancer, as well as to reduce the risk of developing malignant tumors.

#### 2.1.3. Importance of Vitamin D Supplementation in Breast Cancer Treatment

Gnagnarella et al. indicated in their review that data collected from various published studies do not allow for drawing unequivocal conclusions about the influence of vitamin D supplementation on mortality in cancer patients, and more randomized clinical trials are therefore needed [96]. A study by Lin et al. on premenopausal women showed that simultaneous intake of increased doses of calcium and vitamin D in the diet or in the form of dietary supplements reduces the risk of developing breast cancer [42]. This finding was also confirmed in other studies, which demonstrated a significant correlation between a diet rich in calcium and vitamin D and the incidence of ER+ breast cancer [42,43,47]. However, other randomized controlled trials did not show any substantial relationship between the intake of calcium and vitamin D and the risk of developing breast cancer (Table 1 and Table 2) [39,44,97].

Antineoplastic treatment of hormone-dependent breast cancer is based on the use of selective estrogen receptor modulators (SERMs), such as tamoxifen (TAM). However, if used for a long term, TAM may exert an agonistic effect on estrogens, stimulating endometrial tissue to express high levels of ER, thus leading to the development of endometrial diseases including cancer [98,99,100]. In addition, the use of TAM can result in acquired resistance in many patients, which may depend on the presence of ERβ, as well as increased expression of EGFR, HER2, or insulin-like growth factor 1 receptor [101]. Due to the numerous side effects and resistance related to the use of SERMs, the treatment strategy was changed from restricting access to ERs to reducing or blocking estrogen synthesis. Aromatase inhibitors (AIs) were developed to prevent the conversion of androgens to estrogens, by binding to the active site of the enzyme. Currently, apart from SERMs, third-generation AIs, such as anastrozole or letrozole, are used in the treatment of breast cancer. These inhibitors outperform compounds of previous generations in the specificity of binding to the enzyme. They also have a higher efficacy and produce less intrusive side effects. Long-term use of anastrozole, as well as other AIs, leads to almost complete inhibition of estrogen synthesis, thus inhibiting the development of breast cancer. However, AIs also exert undesirable effects on the body. They are not tissue-specific and inhibit the activity of aromatase in each cell where it occurs. Complete elimination of estrogens results in abnormal functioning of the bone tissue, bone decalcification, reduction in bone density, greater bone fragility, and appearance of pain and aggravation of osteoporosis in patients suffering from this condition [102,103]. Since estrogens are also vital for the proper functioning of skeletal and smooth muscles, people treated with AI experience muscle pain due to estrogen deficiency. To overcome the side effects of AI on the skeleton, as well as to reduce the formation of bone metastases, bisphosphonates are used [104,105,106]. These compounds prevent bone resorption and metastasis by inhibiting the adhesion and proliferation of cancer cells in the bone, as well as hypercalcemia associated with cancer development. However, the use of bisphosphonates also causes side effects, including nephrotoxicity and hypophosphatemia [106]. Moreover, studies have demonstrated vitamin D deficiency among breast cancer patients with metastatic bone disease who were treated with bisphosphonates and recommend vitamin D supplementation for these patients [107,108,109,110]. It was also proven that in patients treated with AIs for ER+ breast cancer the proper management of osteoporosis with the use of bisphosphonates and vitamin D allows reducing the risk of relapse and death by 50% [111]. In addition, some studies have indicated that anticancer therapy can be supported with the intake of high doses of vitamin D_3_ and calcium, omitting bisphosphonates, due to their side effects [112,113]. The results of these studies are summarized in Table 2. The use of high concentrations of vitamin D_3_ and calcium seems to be justified due to the fact that the development of breast cancer is accompanied by a decrease in the vitamin D_3_ concentration in the serum [109,114] and its synthesis in the cancer and surrounding tissues [18]. Recently published studies showed that de novo postdiagnosis supplementation of vitamin D (n-5417) caused a reduction in breast cancer-specific mortality [115,116]. Not only patients with ER+ breast cancer and treated with endocrine therapies may benefit from vitamin D. A retrospective review analyzing the effect of trastuzumab treatment in nonmetastatic HER2+ breast cancer patients revealed that high vitamin D intake during therapy improved disease-free survival but not overall survival [117]. The literature also includes interventional studies. The results of a randomized, placebo-controlled trial carried out for 5 years on a large number of participants highlighted that supplementation with 2000 IU of vitamin D had no effect on the incidence of invasive breast cancer [118] (Table 2). But on the other side, a 20 years of follow-up study in cancer-free postmenopausal women with daily supplementation of calcium (1000 mg) and vitamin D (starting from 400 IU/day) was associated with lower risk of Ductal Carcinoma in Situ (DCIS), which may raise the possibility that consequent supplement intake may provide long-term benefits in the prevention of DCIS [119]. Patients with nonmetastatic HER+ breast cancer supplemented with vitamin D during neoadiuvant chemotherapy revealed improved disease-free survival [117]. Another cohort study revealed that everyday supplementation with vitamin D and calcium by menopausal hormone therapy users is associated with decreased postmenopausal breast cancer risk [120].

**Table 2 cancers-14-03649-t002:** Interventional studies in breast cancer patients and healthy volunteers.

Vitamin D Dosage Used for Treatment	Patient Characteristics; Trial Type	Results	References
Healthy people			
2000 IU/d of vitamin D_3_ and 1500 mg/d of calcium for 4 years	2303 healthy postmenopausal women, 55 years or older; 4-year, double-blind, placebo-controlled, population-based randomized clinical trial	Among healthy postmenopausal older women a mean baseline serum 25(OH)D level of 32.8 ng/mL, vitamin D_3_ and calcium supplementation didn’t decrease significantly the risk of all-type cancer in a 4 years study	[97]
2000 IU/d for 5 years	1617 participants (793 in the vitamin D group and 824 in the placebo group); randomized, double-blind, placebo-controlled trial, with a two-by-two factorial design	Treatment did not reduce the incidence of breast cancer	[118]
Breast cancer patients			
4000 IU vitamin D_3_ daily for 12 weeks	168 breast cancer survivors; single-arm nonrandomized before-and-after trial has been registered in the Iranian Registry of Clinical Trials (IRCT) under the identification code: IRCT2017091736244N1	The association between the *VDR* SNPs (*Apa*I, *Taq*I, *Fok*I, *Bsm*I, and *Cdx2*) and changes in response was assessed. *Cdx2* genotypes AA and GA, compared to GG, showed higher plasma levels of MMP9; *Bsm*I bb genotype showed a greater decrease in circulating TNF-α levels after vitamin D_3_ supplementation; *VDR* genetic polymorphisms were not associated with longitudinal changes in the remaining cancer biomarkers. In breast cancer survivors with low 25(OH)D plasma levels and vitamin D3 supplementation changes in certain inflammatory biomarkers may be affected by VDR SNPs and haplotypes	[121]
4000 IU vitamin D_3_ daily for 12 weeks	176 breast cancer survivors who had completed treatment protocol, including surgery, radiotherapy, and chemotherapy; trial has been registered on the IRCT under the identification code: IRCT2017091736244N1	85% of women had insufficient and inadequate levels of plasma 25(OH)D at baseline; aa genotype of *Apa*I showed a greater increase in muscle mass and higher decrease in low-density lipoprotein cholesterol levels; Bb genotype of the *Bsm*I VDR showed a higher increase in waist circumference following vitamin D_3_ supplementation; haplotype score analyses showed a significant association between inferred haplotypes from *Bsm*I, *Apa*I, *Taq*I, and *Fok*I, *Bsm*I and *Cdx2* VDR polymorphisms, and on-study visceral fat changes	[122]
50,000 IU/week for 8 weeks	56 breast cancer patients; 2 treatment arms: placebo and vitamin D_3_ through a 2-month intervention period; double-blind, placebo-controlled trial	Supplementation with vitamin D_3_ increased the total antioxidant capacity (TAC) in breast cancer women; no effect was found on inflammatory markers. Serum TAC in the TT/Tt and Ff genotypes was more responsive to vitamin D supplement compared with FF/ff and tt genotypes	[123]
40,000 IU/d of vitamin D_3_ or placebo for 2–6 weeks prior to breast surgery	120 newly diagnosed breast cancer patients; prospective, randomized, phase 2, double-blinded presurgical window of opportunity trial; trial registration: NCT01948128.	Significantly higher levels of serum 25(OH)D in the vitamin D-treated group were not associated with any significant effects on tumor proliferation and apoptosis	[124]
10,000 IU daily in the interval between biopsy and surgery	29 breast cancer patients; controlled, and blinded trial in women with core needle biopsies positive for breast cancer, but without the presence of metastatic lesions; ancillary study of a breast cancer trial (NCT01472445)	Vitamin D supplementation can decrease circulating 27-hydroxycholesterol in breast cancer patients, likely by CYP27A1 inhibition. This suggests a new and additional modality by which vitamin D can inhibit ER+ breast cancer growth; a larger study is needed for verification	[125]
10,000 IU vitamin D_3_ and 1000 mg calcium each day for 4 months	40 patients with bone metastases treated with bisphosphonates; single-arm, phase 2 study	Treatment was safe, and reduced inappropriately elevated PTH levels caused by long-term bisphosphonate use; no significant palliative benefit or any significant change in bone resorption was observed	[126]
2000 IU/1000 mg and 4000 IU/1000 mg based on baseline serum 25(OH)D for 12 weeks	82 breast cancer patients treated with letrozole	Vitamin D_3_ supplementation significantly improved serum 25(OH)D concentrations and decreased letrozole-induced arthralgia	[113]
50,000 IU/week for 12 weeks	60 breast cancer patients treated with letrozole	Vitamin D_3_ supplementation is safe, and may reduce disability from AI-induced arthralgias	[112]
30,000 IU oral vitamin D_3_/week for 24 weeks	160 women with stage I–III breast cancer starting adjuvant letrozole and 25(OH)D level ≤ 40 ng/mL	Treatment was safe and effective in achieving adequate vitamin D levels, but not associated with a decrease in AI-associated musculoskeletal symptoms	[127]
800 IU/d with calcium but women with baseline 25(OH)D < 30 ng/mL also received 16,000 IU of vitamin D_3_ every 2 weeks for 3 months	290 breast cancer patients starting AI, prospective cohort	40 ng/mL 25(OH)D may prevent development of AI-induced arthralgia but higher loading doses are required to achieve this level in women with deficiency at baseline	[128]

### 2.2. Impact of Vitamin D on Animal Models of Breast Cancer (Pre-Clinical Studies and Signal Transduction Data)

Clinical findings concerning the progression of tumor growth and metastasis via vitamin D deficiency are supported by studies performed using transgenic mouse models of spontaneous mammary gland cancer [129] or xenografts of human breast cancer cell lines [130] and allografted mouse tumors [131,132,133,134]. In addition, a study on a breast cancer bone metastasis model showed that vitamin D deficiency resulting from 1α-hydroxylase (*Cyp27b1*) knockout increased the growth of TM40D mammary gland tumor in bone and accelerated tumor-induced bone destruction [132]. Moreover, in a mouse breast cancer model, targeted *Cyp27b1* gene ablation in the mammary epithelium of polyoma middle T antigen-mouse mammary tumor virus (PyMT-MMTV) led to the initiation and acceleration of spontaneous mammary tumorigenesis [135]. The anticancer and antimetastatic activity of calcitriol and its analogs have been observed in various breast cancer models [136,137,138]. Furthermore, VDR knockdown was found to significantly accelerate the metastasis of MDA-MB-231 human breast cancer cells to bone [139], as well as increase primary tumor growth and metastasis of 168FARN mouse mammary gland tumors to the liver [131]. Additionally, the loss of VDR signaling in MMTV-Ron VDR^−^/^−^ mice caused an increase in spontaneous breast tumorigenesis and enhanced metastasis to the lungs and liver [140]. On the other hand, Trivedi et al. indicated that ablation of VDR (in the absence of ligand) reduced MCF-7 tumor growth in mammary fat pad and in bone [141]. Furthermore, the authors observed protumoral [142] or prometastatic activity of calcitriol in prostate [143] and breast cancer models [144,145].

Experimental studies also show that combined use of vitamin D or its analogs with chemotherapy agents has potential benefits. As reported by Krishnan et al. in a MCF-7 xenograft study the aromatase mRNA was reduced in the tumor and surrounding mammary adipose tissue after calcitriol treatment, but there was no change in aromatase mRNA noticed in the ovary. This selective aromatase gene regulation by calcitriol results from/is dependent on the presence of different promoters in these tissues. Calcitriol acts through the vitamin D response element present in the aromatase promoter II and, on the other way, suppresses the aromatase gene transcription by reducing the level of prostaglandins involved in estrogen synthesis in the breast tissue [146].

Thus, combined use of calcitriol with one of three different AIs, namely exemestane, letrozole, or anastrozole, caused enhanced inhibition of MCF-7 cell growth [146]. Our studies confirmed these findings in human MCF-7 xenografted mice demonstrating tumor growth inhibition after treatment with new vitamin D analogs, PRI-2191 and PRI-2205, either alone or in combination with anastrozole [147]. In addition, Swami et al. revealed that calcitriol acted as a selective aromatase modulator in mice, decreasing aromatase expression in the mammary adipose tissue and increasing it in bone marrow cells, but not altering the enzyme expression in the ovaries and uteri [148]. Some studies have shown that the EB1089 analog, when used in combination with low doses of clinically used AIs, also decreased the expression and activity of aromatase, which led to the growth inhibition of breast cancer cell lines [149].

Other in vivo studies that have analyzed the use of vitamin D or its metabolites and analogs in breast cancer treatment, either alone or in combination with other anticancer treatment strategies, are summarized in Table 3. A recent review extensively discussed the use of calcitriol and its analogs in combination with other therapeutic strategies in breast cancer [150].

## 3. Role of Th17 Cells in Breast Cancer

After activation, CD4+ T lymphocytes, which are the central regulatory cells of innate and adaptive immunity, differentiate into various T helper (Th) subsets to ensure homeostasis. Among these subsets, the well-known ones are Th1 cells producing interferon-γ (IFN-γ) and Th2 producing interleukin (IL) 4. The other identified Th subset is Th17 producing IL-17. CD4+ T cells also differentiate into T regulatory cells (Treg) expressing forkhead box P3 (FOXP3) [173] (Figure 1). Th17 cells are the main source of the proinflammatory cytokine IL-17; however, the receptor of this cytokine (IL-17R) is distributed ubiquitously. IL-17 (IL-17A) is a member of the IL-17 cytokine family consisting of IL-17A–F (IL-17E is also known as IL-25) and directly links inflammatory responses and T-cell activation [174,175].

As transforming growth factor β (TGF-β) plays an important role in the differentiation of both Th17 and Treg cells (Figure 1), IL-6 counteracts the differentiation of Treg cells upon TGF-β and directs the differentiation of Th17 [176,177]. It has been shown that IL-6 upregulates the expression of IL-21 by activating signal transducer and activator of transcription 3 (STAT3), causing an increase in the expression of retinoid-acid receptor-related orphan nuclear receptor (ROR) γt, RORα, and IL-23R and ultimately promoting the complete differentiation of Th17 cells. On the other hand, STAT3 deficiency impaired RORγt expression and elevated the expression of T-bet (member of T-box family transcription factors) and FOXP3 [178]. RORγt belongs to the retinol family and regulates the differentiation of Th17 cells, while RORα promotes the differentiation of these cells. RORγt and RORα can synergistically induce Th17 differentiation [179]. IL-21 produced by Th17 cells stimulates their autocrine formation [180]. IL-1 receptor type 1 (Il1r1) gene is also the promoter of Th17 cell differentiation [181]. Moreover, Th17 cells express CD39 and CD73 ectonucleotidases, leading to the release of adenosine and the suppression of effector T cells.

As Chalmin et al. have shown, during the differentiation of Th17 cells, the ectonucleotidase expression is transcriptionally regulated by IL-6 (STAT3 activation) and by TGF-β-mediated downregulation of zinc finger protein growth factor independent-1 (*Gfi-1*). The expression of CD39 ectonucleotidase in the case of Th17 cells determines their immunosuppressive nature in cancer [182].

In vitro induction of Th17 cells can be achieved directly with the use of anti-CD3/CD28 antibodies and cytokines. However, in vivo priming of Th17 cells requires dendritic cells (DCs) that present antigen, provide costimulatory signals, and help in the synthesis of IL-1, IL-6, TGF-β, tumor necrosis factor (TNF) α, and IL-23 cytokines [183,184,185]. It is also known that fibroblasts support IL-23 secretion from DCs that are preactivated by lipopolysaccharide compared to DCs activated by lipopolysaccharide alone. It is realized via a complex feedback-loop mechanism involving IL-1β/TNF-α (from activated DCs), which stimulates prostaglandin E_2_ (PGE2) production by fibroblasts. PGE2, in turn, acts on activated DCs and increases the release of IL-23 from these cells. Furthermore, compared to DCs alone, fibroblast-stimulated DCs performed better in promoting the expansion of Th17 cells in a cyclooxygenase (COX)-2-, IL-23-dependent manner [186] (Figure 2). A recent review by Pastor-Fernandez et al. broadly described the role of IL-23 in the differentiation of Th17 cells [187]. The role of IL-33 in maintaining the balance between Treg and Th17 cells has also been emphasized. DCs matured upon IL-33 inhibited the differentiation of CD4+ Treg cells in favor of Th17, which was realized through IL-6 signaling [188].

Recent studies suggest that osteopontin (OPN) is also required for DCs to induce Th17 differentiation [189] and IL-17 production [190]. In addition, studies on acute coronary syndrome showed that OPN is involved in inflammation through its direct effect on IL-17-producing cells [191]. Moreover, the direct effect of OPN on the differentiation of Th17 cells is exerted through interaction of these cells with its receptors [192]. OPN contains an Arg-Gly-Asp (RGD) sequence, which is common to many extracellular matrix proteins and mediates the interaction of OPN with multiple integrins such as αvβ1, αvβ3, αvβ5, and α5β1 [193]. CD44, which is another important receptor of OPN, is involved in T cell activation [194]. Studies have shown that blocking of CD44 and, to a lesser extent, β1 integrin subunit (CD29), resulted in a significant reduction in Th17 cell differentiation, while the addition of a CD51 (integrin αv)-blocking antibody did not result in such effect, indicating that the effect of OPN was mediated through CD44 and CD29. Furthermore, the production of IL-17 from OPN-stimulated CD4+ T cells was inhibited by CD44 or CD29 antibodies in a dose-dependent manner [195]. Other studies have also pointed out the role of CD61 (β3 integrin), another OPN receptor, in Th17 cell differentiation [192].

Estradiol (E2) is also a factor that affects Th17 cells. Th17 cells express both ERα and ERβ. Studies have reported various effects of E2 on Th17 cell differentiation. In mouse splenocytes, E2 inhibits Th17 cell differentiation and IL-17 production by inhibiting the expression of RORγt [196]. Similarly, in E2 deficiency-induced bone loss, the differentiation of Th17 cells was increased, accompanied by upregulation of STAT3, RORγt, and RORα and downregulation of FOXP3 [197]. However, studies performed by Andersson et al. on in experimental autoimmune arthritis (AA) have shown that estradiol treatment increases the amount of Th17 cells in lymph nodes during the early stage of arthritis development. In the advanced stage of the disease, estradiol acts in the opposite way, diminishing the number of Th17 cells in joints. The authors of the studies suggest the observed effect of estradiol action may be caused by the interference of E2 with CCR6-CCL20 (C-C chemokine receptor 6–C-C motif chemokine ligand 20) pathway, which is important for the migration of Th17 cells. E2 increased the expression of CCR6 on Th17 cells in lymph nodes as well as the expression of the corresponding CCL20 within lymph nodes [198]. Other studies on mice splenocytes demonstrated that ERα signaling increased IL-17A production in Th17 cells by upregulating the expression of IL-23R and promoting mitochondrial respiration and proliferation [199]. Deletion of ERα, but not ERβ, caused a significant decline in the production of IL-17A and surface expression of IL-23R on Th17 cells. These effects are realized through an increase in the relative expression of Let7f microRNA in Th17 cells. The findings of these studies show that ERα signaling regulates Th17 cell differentiation by influencing the Let7f/IL-23R pathway [200,201]. Thus, it seems that the influence of E2 on Th17 cells may depend on the environment in which the study is conducted and the type of disease analyzed, and the surrounding environment moderates the direction of E2′s influence on these cells.

It is known that the subset of Th17 cells is transient in nature. For example, in specific experimental conditions, CD4+ T cells may exhibit diminished IL-17 expression and upregulated IFN-γ expression [202]. In addition, Th17 lineage exhibits plasticity and can transdifferentiate into Treg cells [203]. FOXP3-expressing Treg cells, mostly represented by CD4+ T cells that express CD25 (IL-2 receptor α-chain), are important for controlling self-tolerance and immune homeostasis, but also suppress antitumor immune responses and favor tumor progression [204]. Th17 cells can act as a source of tumor-induced FOXP3+ cells, as it was shown by Downs-Canner et al. In addition to natural Treg and induced Treg cells formed from naïve precursors, suppressive IL-17A+ FOXP3+ and ex-Th17 FOXP3+ cells are converted from IL-17A+ FOXP3− cells in tumor-bearing mice [204]. Moreover, Th17/Th1 cells produce both IL-17 and IFNγ and Th17 cells stimulated by IL-12 can shift to Th17/Th1 [205,206]. Activated Th17 cells can produce IL-22 along with IL-17 [207], and represent a distinct population apart from Th22 cells [208]. Intestinal Th17 cells can differentiate into T follicular helper (Tfh) cells in Peyer’s patches [209]. The plasticity of Th17 cells has been described in detail elsewhere [210,211,212,213] (Figure 3).

The phosphoinositide 3-kinase (PI3K)/serine/threonine-protein kinase (AKT) signaling pathway is involved in the processes of cell growth, differentiation, and apoptosis, and its activation is critical for the completion of cell cycle and cell differentiation. In addition, T cells proliferation and migration is also regulated by PI3K/AKT pathway. It has been shown that Th17 cell differentiation (both in vitro and in vivo) can be regulated by mTORC1 and mTORC2, mammalian targets of rapamycin (mTOR) complexes (via PI3K/AKT in different ways) [192,214]. Activation of PI3K and/or mTORC1 enhances the Th17 cell differentiation, but on the other hand, the inhibition of PI3K and/or mTORC1 in CD4+ T cells causes an increase in the differentiation of Treg cells [215].

Th17 cells can play both protective and pathogenic roles in immunity. The protective action of these effectors is related to the suppression of pathogens including *Candida albicans* and *Staphylococcus aureus*. However, it is believed that Th17 cells also induce inflammation and tissue damage [216].

### Importance of Th17 Cells in Breast Cancer

A number of studies have confirmed the presence of Th17 cells in various types of cancers (e.g., breast, ovarian, colorectal, cervical cancer, and melanoma) and the significance of these cells in these diseases [217,218,219]. However, it is difficult to provide a clear description of the role played by Th17 cells in tumor development due to complex interactions occurring between cancer cells and the components of the host microenvironment [213]. Inflammation is often associated with cancer progression and actively contributes to the survival of cancer cells, angiogenesis, and metastasis [220]. It is known that tumor cells and cancer-associated fibroblasts (CAFs) create an inflammatory environment favorable for the recruitment of Th17 cells [221].

Although many studies have been carried out on Th17 cells, the role of these cells in breast cancer remains undefined [222]. Nevertheless, the majority of evidence indicates that Th17 cells exhibit prooncogenic properties in breast cancer. Clinical analyses have shown that the level of Th17 cells/IL-17 cytokine is usually altered in patients with breast cancer [217,223]. Compared to healthy donors, the level of Th17 cells is higher in the blood of breast cancer patients and correlates with elevated levels of C-X-C motif chemokine ligand (CXCL) 1. CXCL1, a proinflammatory chemokine produced by breast cancer cells, can promote cancer growth and development [224]. A positive correlation between the levels of IL-17 and macrophage infiltration inhibitory factor (MIF) has also been observed, and both IL-17 and MIF were linked with a high risk of developing breast cancer of aggressive molecular subtypes [225]. In breast tumors characterized by matrix metalloprotease (MMP) 11 expression by intratumoral mononuclear inflammatory cells, the level of expression of inflammatory factors, associated with distant metastasis development, such as IL-17 and NFκB (nuclear factor kappa-light-chain-enhancer of activated B cells), was shown to be significantly higher [226]. Infiltration by IL-17+ T cells in TNBC patients was associated with a poor recurrence-free survival [227]. Moreover, infiltration by Th17 cells is preferably observed in ER−, PR−, and TNBC tumors and is a poor prognostic factor [228]; it also correlates with failure of complete pathological response [229,230]. Another study demonstrated that an increased number of IL-17A-producing cells are found mainly in ER– and triple-negative/basal-like breast tumors [231]. On the other hand, high levels of ER suppress Th17 cell infiltration and IL-17 signal transduction, causing a reduction in PD-1/PD-L1 expression and CD8+ T cell infiltration in breast cancer [232]. In turn, a study by Horlock et al. showed that the number of circulating Th17 cells was the lowest in patients with HER2+ breast cancer compared to healthy controls and HER2– patients. An inverse relationship was also observed between the frequencies of Treg and Th17 cells in metastatic breast cancer with a significant reduction in the level of Treg cells during treatment with trastuzumab, whereas the level of Th17 cells was concomitantly increased [233]. On the other hand, a study investigated the distribution of IL-17-producing CD4+ T-cells in relation to Treg cells in tumor-infiltrating lymphocytes (TILs) and peripheral blood mononuclear cells (PBMCs) collected from breast cancer patients. The frequency of Th17 cells was found to be significantly higher in TILs than in PBMCs obtained from early breast cancer patients. In the TILs collected from advanced breast cancer patients, the frequency of Th17 cells was also significantly higher compared to that in PBMCs but lower compared to PBMCs from patients with early disease. Based on these findings, the authors concluded that the accumulation of Th17 and Treg cells in the tumor microenvironment of breast cancer occurred during the early stage of the disease. It was also indicated that Th17 cell infiltration gradually decreased but Treg cells continued to accumulate as the disease progressed [234]. However, studies on mice showed that the Th17 subpopulation was dominant in CD4+ T cells from TILs, and the population was also higher in the late tumor stages [235].

A meta-analysis of IL-17A estimation by immunohistochemistry, overall survival, and disease-free survival in patients with solid tumors indicated that in most of the cases these parameters were worse with higher levels of IL-17A. IL-17A was also associated with an advanced stage of cancer [222]. However, the association between the level of intratumoral Th17 cells and blood level of IL-17 was not clear. In addition, their effects did not seem to be unequivocal; thus, when Th17 cytokines (IL-17A and IL-17F), which were upregulated in TNBC, specifically in T cell noninflamed tumors, were exploited by the METABRIC transcriptomic dataset, a high expression of Th17 metagene was identified as an indicator of good prognosis of T cell noninflamed TNBC [236].

In addition to IL-17A, other IL-17 family members and their receptors have been analyzed in breast cancer patients. Mombelli et al. reported that mRNA expression of IL-17A and IL-17E receptor subunits was upregulated in breast cancers in comparison to normal samples. Furthermore, it seems that IL-17E, which is usually undetectable in normal breasts, is overexpressed in cancerous tissues [237]. It can also promote resistance to antimitotic and anti-EGFR therapies. In the breast cancer cell lines IJG-1731, BT20, and MDA-MB-468, EGFR phosphorylation is stimulated by epidermal growth factor and IL-17E. IL-17E also activates kinases that are crucial for EGFR signaling, such as PYK-2, Src, and STAT3 [238]. IL-17E and IL-17A induce cell proliferation and survival by activating pathways including c-RAF, ERK1/2, and p70S6 kinase, which also leads to docetaxel resistance [237]. Cochaud et al. authenticated that in human breast cancer cell lines recombinant IL-17A recruits the mitogen-activated protein kinases (MAPK) pathway by upregulating phosphorylated ERK1/2 which results in stimulation of cell proliferation, migration and invasion, and resistance to commonly used chemotherapeutic agents such as docetaxel [231].

A high level of IL-17B was found in patient biopsies, which was associated with a decrease in overall survival and with poor prognosis. Moreover, overexpression of IL-17RB was associated with reduced disease-free survival. Both overall and disease-free survival were reduced in patients with overexpression of IL-17B and IL-17RB [239]. In another study, the overexpression of IL-17RA and IL-17RB is associated with poor prognosis and shorter survival rate [240]. In breast cancer cell lines BT20, MDA-MB-468, and MCF-7, IL-17B induced resistance to paclitaxel, and activation of the extracellular signal-regulated kinase 1/2 (ERK1/2) pathway, leading to the upregulation of B-cell lymphoma 2 (Bcl-2) [239]. Moreover, IL-17RB and IL-17B amplification can promote tumorigenicity in breast cancer via the activation of NF-kB and Bcl-2. It has been shown that depletion of IL-17RB in tastuzumab-resistant cell lines ceased colony formation and retarded tumor growth in mice [240]. Recently, Bastid et al. comprehensively described the role of IL-17B and IL-17RB signaling pathways in cancer [241].

Kiyomi et al. reported that tumor tissues resected from breast cancer patients produced Th17 cytokines when cultured in three-dimensional gelatin polymer culture system [242]. Tumor cells and CAFs produce microenvironmental factors, such as RANTES (Regulated on Activation, Normal T-cell Expressed and Secreted) and monocyte chemoattractant protein-1 (MCP-1/CCL-2) chemoattractants, which mediate the recruitment of Th17 cells, and IL-23 and TGF-β, which are important factors of Th17 cell differentiation and generation. They also allow cell contact, inducing the generation and expansion of Th17 cells. Colony of Th17 cells in TILs obtained from patients produced high levels of IL-8, IL-17, and TNF and a low level of IL-6. Th17 clones expressed chemokine receptors such as CCR2, CCR4, CCR5, CCR6, CCR7, and CXC chemokine receptor (CXCR) 3, which are homeostatic chemokine receptors as well as trafficking receptors found commonly in other T cell lineages, including Treg cells [221]. Tumor production of IL-6 and TGF-β stimulated the differentiation of Th17 cells into CD25^high^/CD39/CD73 Th17 cells. Th17 CD25^high^ cells accumulate in breast cancer tissue by recruitment via CCL20/CCR6. Intratumoral Th17 cells, which are also known as memory CD25^high^/CCR6+ Th17 cells, express IL-17, RORγ, FOXP3, CD39, and CD73. CD39 and CD73 are ectonucleotidases, which catalyze the transformation of ATP, and can lower T cell response. When these enzymes accumulate, they can weaken T cell immunity in breast cancer patients by suppressing CD4+ and CD8+ T cells, which worsens relapse-free and overall survival [243]. In blood samples and invasive ductal carcinoma (IDC) tissue collected from breast cancer patients, Th17-related molecules (IL-17A, RORC, and CCR6), produced by tumor-infiltrating CD4+ and CD8+ T lymphocytes, were observed to be upregulated. Angiogenic factors CXCL8, MMP-2, and MMP-9 and vascular endothelial growth factor (VEGF)-A were detected within the tumor and shown to be induced by IL-17, which correlated with poor prognosis. The accumulation of Treg and Th17 cells within an invasive breast tumor may promote the growth and survival of the tumor cells, and the presence of Treg cells and high levels of TGF-β may also favor the development of Th17 cells [244].

The genetic factors involved in the regulation of Th17 cell differentiation are currently being investigated. Numerous long noncoding RNAs (lncRNAs) are reported to regulate immune response in breast cancer patients [245]. One epigenetically dysregulated lncRNA (LINC01983) and four lncRNA regulators (UCA1, RP11-221J22.2, RP11-221J22.1, and RP1-212P9.3) were identified to act as prognostic biomarkers of luminal breast cancer by controlling the TNF signaling pathway, Th17 cell differentiation, and T cell migration [246].

Single-nucleotide polymorphisms (SNPs) of the IL-17 gene have been shown to be correlated with susceptibility to cancer [247]. Wang et al. analyzed SNPs of IL-17A and F genes and reported that rs2275913 polymorphism of IL-17A gene was associated with an increased risk of breast cancer in Chinese women [248]. However, Naeimi et al. indicated that polymorphisms of IL-17A and IL-17F genes have no significance in the susceptibility of women from southern Iran to breast cancer [249].

In addition, studies on animal models showed protumoral, prometastatic, and proangiogenic activity of IL-17 as well as the impact of this molecule on chemoresistance. A number of studies were conducted in the 4T1 mouse mammary gland tumor model, the growth of which is associated with high immune response including large leukocytosis, and lung and tumor infiltration by neutrophils [250]. Th17 lymphocytes were shown to be increased in the peripheral blood, spleen, and tumor tissue of 4T1 tumor-bearing mice [251,252]. PGE2 secreted by this tumor induced the production of IL-23 in the tumor microenvironment, leading to the expansion of Th17 cells [252]. In another study, the authors characterized T cells specific for 4T1 cancer and described them as receptor activator for nuclear factor κB ligand (RANKL)+ IL-17F+ CD4+ T cells [253]. Such cells arrive in the bone marrow before metastatic cells and build a premetastatic niche, which in effect leads to premetastatic osteolytic disease and bone metastases. 4T1-conditioned media support the differentiation of DCs to mature and activated multinucleated giant cells expressing TRAP and IL-23. These cytokines are involved in the activation of 4T1 tumor-specific T cells determined by RANKL and IL-17 production [254]. Moreover, the production of IL-17F and RANKL was only observed in cells derived from mice bearing 4T1 metastatic tumors, and not in cells from mice bearing 67NR nonmetastatic cells [254]. Administration of IL-17 in 4T1 tumor-bearing mice resulted in an increase in tumor size and a higher microvascular density [251]. Furthermore, a decrease in the levels of IL-17A caused by treatment with endothelin-1 receptor dual antagonist led to the slowdown of the growth of 4T1 tumor. In immunocompetent mice implanted with 4T1 cells, such treatment resulted in a reduced tumor growth and a decrease in the concentrations of proinflammatory TNF-α and IL-17 cytokines [255]. Similarly, knockdown of IL-17R in 4T1 mouse mammary gland cancer cells caused a reduction in tumor size and enhanced apoptosis [256]. Inhibition of IL-17 significantly reduced the metastases of spontaneously developing mammary gland carcinoma in MMTV-PyV MT mice with induced AA. In these mice, AA as well as lung and bone metastasis correlated with a high level of IL-17 [257]. In MCF-7, MDA-MB-157, MDA-MB-361, and MDA-MB-468 human breast cancer cell lines, high levels of IL-17RB as well as high IL-17RB mRNA expression have been observed. Depletion of IL-17RB resulted in inhibited colony formation and retarded MDA-MB-361 tumor growth in mice [240]. Inhibition of IL-17 also reduced the proliferation and colony formation as well as tumor growth, as revealed by chorioallantoic membrane assay (CAM) using MCF-7 cells [258]. At the same time, when stimulated with Th17 cells, MCF-7, MDA-MB-435, T47D, and MDA-MB 231 cells showed increased matrigel invasion [259].

In a murine study on mice bearing parental Cl66 murine mammary tumors and Cl66 cells resistant to doxorubicin (Cl66-Dox) or paclitaxel (Cl66-Pac) Wu et al. revealed the role of IL-17, CXCR2 ligands, and cancer-associated neutrophils in chemotherapy resistance and metastasis of breast cancer [260]. In tumor tissue of resistant models increased levels of IL-17R, CXCR2 chemokines, and CXCR2 were observed in comparison to C166 tumor tissue. What speaks for the significance of Th17 cells in chemoresistant cancer cells is the higher infiltration grade by Th17 and neutrophils in C166-Dox and C166-Pac models.

In addition, CD8+ T cells (splenocytes) from 4T1 tumor-bearing mice expressed IL-17, which promoted cell survival and reduced apoptosis. Addition of TGF-β and IL-6 caused a threefold higher IL-17 expression in CD8+ T cells from tumor-bearing mice than from naïve mice. A significant decrease in tumor size was also noted after blocking TGF-β and after depletion of CD8+ T lymphocytes. A similar reduction effect was observed on lung metastases [256].

It was shown that IL-17-producing γδ T cells and neutrophils synergistically promoted breast cancer metastasis in the mouse model of spontaneous metastasis [261]. IL-17+ γδT cells played an important role in oxidative metabolism, with increased mitochondrial mass and activity. Protumoral IL-17+ γδT cells selectively showed high lipid uptake and intracellular lipid storage and expanded in the tumors of obese mice [262].

IL-17E was also found to exhibit antitumor effects in mice lacking various T lymphocytes-bearing tumors, including breast cancer, but not in mice lacking both T and B lymphocytes. Treatment with IL-17E resulted in a significant increase in IL-5 serum levels and increased numbers of eosinophils in peripheral blood of tumor bearing mice. Also a significant increase in eosinophils was observed in spleens isolated from IL-17E-treated mice, which correlated with the antitumor activity of IL-17E in a dose-dependent manner. Moreover, B cells play also an important role in IL-17E-mediated antitumor activity. IL-17E activated signaling pathways in B cells in vitro [263]. In addition, breast cancer cells treated with IL-17E obtained from nonmalignant mammary epithelial cells-conditioned medium showed decreased colony formation [264]. Myeloid-derived suppressor cells (MDSCs), which are found at increased levels in breast cancer patients, were purified from mice bearing MCF-7 tumors and treated with IL-17. This treatment significantly induced the differentiation of MDSCs, inhibited their proliferation, and triggered apoptosis as well as inhibited the activation of STAT3 in these cells (Ma, Huang, and Kong, 2018).

The pro- and anti-cancer effect of Th17 cells and their cytokines has been reviewed by Fabre et al. [265] and Qianmeni et al. [266] and is summarized in Figure 4.

## 4. Action of Vitamin D on the Immune System in Cancer

Vitamin D may affect the growth and progression of tumors by directly influencing tumor cells or their microenvironment. Figure 5 summarizes the direct effects of vitamin D compounds observed on breast cancer cells [267,268,269]. The tumor microenvironment, which includes vascular endothelial cells, immune cells, and fibroblasts, is an important element affecting the progression, metastasis, and sensitivity of cancer to clinical therapies. Vitamin D, and its hormonally active form—calcitriol, can affect almost every cell in the body, including the tumor-building cells [269].

CAFs are the main component of cancer stroma. Studies indicate that CAFs promote the development of tumors; however, evidence also suggests that these cells have tumor-suppressing effects [270]. A gene expression study, performed on human CAFs isolated from tumor biopsies of five breast cancer patients, identified a total of 123 genes that are regulated by calcitriol (100 nM). The identified genes include *NRG1* (neuregulin 1), *WNT5A* (Wnt family member 5A), *PDGFC* (platelet-derived growth factor C), and other genes promoting proliferation, which were downregulated in the CAFs, and genes involved in immune modulation such as *NFKBIA* (NFκB inhibitor α) and *TREM-1* (triggering receptor expressed on myeloid cells 1) which were upregulated, as well as *DUSP1* (dual specificity phosphatase 1; a phosphatase that inactivates MAPKs) which was also upregulated. In paired normal fibroblasts, calcitriol modulated the expression of 126 genes (55% of them were also regulated by calcitriol in CAFs), including a few genes involved in proliferation, apoptosis, and differentiation processes [271,272], which were upregulated.

Apart from suppressing VEGF expression and hypoxia inducible factor 1α (HIF-1α) signaling in breast cancer cells [273], studies analyzing the effects of vitamin D on tumor angiogenic processes revealed the direct effects of calcitriol or its analogs on tumor-derived endothelial cells [274] or VEGF-induced bovine aortic endothelial cells [162]. In addition, in vivo studies on nude mice xenotransplanted with human breast cancer cell lines showed weaker vascularization of tumors upon calcitriol treatment [162]. Our studies on 4T1 mammary gland cancer model showed that calcitriol and its two analogs, PRI-2191 (tacalcitol) and PRI-2205, increased blood flow in tumors growing in young mice [144], whereas no such effect was observed in aged mice [151]. In general, vitamin D and its derivatives exert an intense effect on vascular endothelial cells, which is not just observed in the case of neoplastic diseases, as detailed in an interesting review by Kim et al. [275].

Adipose stroma plays an important role in the progression of breast cancer. Especially in obese individuals, the adipose tissue is inflamed, which leads to an unfavorable microenvironment, and along with increased estrogen production, promotes the progression of breast cancer [276]. In their study on the model of MMTV-Wnt1 mouse mammary gland cancer transplanted into ovariectomized (OVX) mice, Williams et al. observed that in mice on high-fat diet a vitamin D-supplemented diet or calcitriol injections slowed down tumor growth and influenced the pathways dysregulated by obesity in both tumor cells and the surrounding breast adipose tissue. Among others, vitamin D suppressed estrogen synthesis and signaling as well as leptin signaling, enhanced phosphorylation of AMP-activated protein kinase (pAMPK) and adiponectin signaling which were dysregulated in response to diet-induced obesity in both breast tumor cells and the surrounding adipose tissue [159,268]. On the other hand, Karkeni et al. performed a study on E0771 mouse mammary gland tumors and observed that gavage with cholecalciferol decreased or increased tumor growth and metastasis in normal and obese mice, respectively [152]. These authors reported that the effect of vitamin D on the studied inflammatory markers was similar in normal and obese mice: vitamin D decreased the mRNA expression of *Il6*, *Ccl5*, and *Cx3cl1* in isolated adipocytes. In the adipocytes of normal mice, the mRNA levels of adiponectin, peroxisome proliferator-activated receptor γ (*Pparg*), PPARγ coactivator 1α (*Pgc1a*), and CCAAT enhancer-binding protein α (*Cebpa*) were also decreased. However, the infiltration of tumor tissue by CD8+ cells was decreased in obese mice, whereas an increased level of these cells was observed in normal mice [152]. Moreover, in normal mice bearing E0771 tumors and treated with cholecalciferol, the spleen and lymph nodes showed decreased infiltration by M1 (F4.80+ CD11b+) macrophages.

Vitamin D has immunosuppressive properties, which, as suggested by some authors, may be beneficial for the treatment of cancers [277]. However, some authors argue that these immunosuppressive effects of vitamin D may adversely affect the efficacy of cancer therapies [278]. Immune cells express VDR and can metabolize vitamin D. In lymphocytes, the expression of VDR is induced after their activation, while DCs and macrophages constitutively express this receptor. These findings indicate the principal role of vitamin D in the modulation of immune and inflammatory responses [278,279,280]. On the other hand, the immune response may be affected by vitamin D direct actions on breast cancer cells. Natural killer (NK) cells form the first line of defense of the innate immune system, but also exhibit direct and indirect antitumor effects through cytotoxic and immune-regulatory properties. In MDA-MB-231 and MCF-7 breast cancer model calcitriol supports NK cells to fight against breast cancer cells partly by reducing miR-302c and miR-520c expression in breast cancer cells and consequently upregulating the activating receptor natural killer group 2, member D (NKG2D) ligands, major histocompatibility complex class I chain-related proteins A and B (MICA/B), and unique long UL16-binding protein 2 (ULBP2) [281].

In our studies on the 4T1 murine mammary gland tumor model, calcitriol and its two low calcemic analogs, PRI-2191 and PRI-2205, increased the metastatic spread in tumor bearing young mice [144,145] while in aged, OVX mice, they revealed transient antimetastatic effects [151] (Table 3). 4T1 tumor cells are not sensitive to proliferation inhibition by calcitriol or its analogs in vitro, and our studies indicated that the growth of 4T1 primary tumor was not affected by treatment with these compounds [144,151]. Other authors have shown that calcitriol stimulated the growth of 4T1 primary tumor [142]. These findings indicate that the prometastatic [282] or protumoral [142] effect of calcitriol is mediated through its impact on immune response, particularly the prevalence of Th2 and Treg cells [142,282]. Moreover, in our studies on mice bearing 4T1 mouse mammary gland cancer, administration of calcitriol or its analogs increased the percentage of Ly6C^low^ anti-inflammatory monocytes in the spleen of young mice. An opposite effect was observed in aged OVX mice [283]. Further in vitro studies were performed on murine bone marrow-derived macrophages (BMDMs) which differentiated into M0, M1, or M2 in the presence or absence of conditioned media from 4T1 (metastatic), 67NR (nonmetastatic), and Eph4-Ev (normal) mouse mammary gland cells under the influence of calcitriol. The results showed that calcitriol enhanced the differentiation of M2 macrophages (increased *Cd206* and *Spp1* (OPN) mRNA expression and CD36, Arg, and CCL2 protein level in M2 BMDMs and decreased *Cd80* and *Spp1* mRNA expression and IL-1, IL-6, OPN, and iNOS protein in M1 BMDMs). 4T1-conditioned media showed a higher effect on gene and protein expression in macrophages after calcitriol treatment, with the greatest effect observed on M2 cells, when compared to 67NR- and Eph4-Ev-conditioned media. This resulted in increased differentiation and properties characteristic of alternative macrophages. Moreover, calcitriol differentiated M2 macrophages stimulated the migration of 4T1 cells through fibronectin [284] (Figure 6).

These findings are important considering the evidence that through various molecules released by cancer cells, such as chemokine CCL2, peripheral monocytes and local macrophages are recruited to the primary tumor and transformed into tumor-associated macrophages [285]. M1 classical macrophages exhibit antitumor activity. However, during cancer progression, a predominance cell similar to M2 alternative macrophages with immunosuppressive properties is observed. These macrophages support the growth of primary tumor, increase the metastatic potential, promote vascularization and remodeling of tumor stroma [286]. In breast cancer, similarly to other cancers, a high infiltration rate by tumor-associated macrophages is associated with poor prognosis [287]. It was shown that in breast cancer patients the disease-free survival and overall survival correlated with the M2 markers expression, namely CD163, CD204, or CD206 [288].

Figure 6 summarizes the published data concerning the impact of vitamin D on breast cancer stromal cells.

## 5. Vitamin D and Th17 Cells

VDR is highly expressed by Th17 cells; calcitriol can modulate the expression of IL-17A in both mouse and human T lymphocytes. Most studies suggest that calcitriol decreases the recruitment of Th17 cells and secretion of IL-17 through the VDR-mediated pathway [289,290,291,292,293]. However, some studies report no correlation between Th17 cells and the level of circulating 25(OH)D, even after supplementation with vitamin D_3_. In a multiple sclerosis study, no correlation was observed between vitamin D_3_ status and individual T cell populations in patients (no supplementation at the beginning) despite high level of vitamin D_3_. Similar results were observed after 12 weeks of vitamin D_3_ supplementation [294,295]. There may be several reasons for these conflicting results. All together; the vitamin D_3_ status in the body, the doses used, and the level of E2, which takes part in *Vdr* gene expression in Th17 cells, play a significant role [289].

The inhibition of proinflammatory Th17 cells as a consequence of calcitriol treatment was observed in a study on young mice with experimental autoimmune encephalomyelitis (EAE) and was defined as transcriptional repression, mediated by the VDR [296]. Moreover, Chang et al. discovered a post-transcriptional mechanism of Th17 cytokines inhibition in EAE by vitamin D, namely induction of the expression of C/EBP homologous protein (CHOP) [297]. The inflammatory environment that accompanies the improper balance between Th17/Treg cells could be regulated by vitamin D even in patients with unexplained recurrent pregnancy loss (URPL) [298]. Vitamin D regulates the expression of genes related to Th17 and Treg cells, increasing the percentage of Treg cells and expression of *FOXP3* gene while diminishing the percentage of Th17 cells and expression of *RORγt* in women with URPL who received vitamin D supplementation for 2 months [299]. In another study in which PBMCs from URPL patients and healthy controls were treated with vitamin D ex vivo, the expression of *FOXP3* and *GITR* genes, and the ratio of *FOXP3*/*RORγt*, were found to be increased [300]. Proper maternal supplementation of vitamin D could attenuate the immune system side effects caused by Bisphenol A (BPA), which is widely found in materials used on a daily basis. Wang et al. showed that exposure of mothers to BPA increased the proliferation of spleen Th17 cells and the serum level of IL-17 in mice offspring. However, vitamin D_3_ supplementation in mothers ameliorated the effects of BPA on the immune system. For instance, it attenuated the upregulation of Th17 proliferation and the expression of RORγt, IL-17, IL-6, and IL-23 in the offspring [301].

Expression of Th2 and inhibition of Th17 polarization through calcitriol supplementation seem to play an important role in suppressing bone destruction induced by periodontitis. Upregulated expression of Treg/Th2-related cytokines (IL-10, IL-4) as well as decreased level of Th1/Th17-related cytokines (such as IFN-γ and IL-17) and proinflammatory immune cells including Th17 was found to be caused by calcitriol in the animal model of periodontitis [302]. In vitro studies revealed that in response to calcitriol administration, Th cells in an inflammatory environment exhibited an enhanced potential for Th2 polarization along with a decreased potential for Th17 polarization in the presence of DCs. Furthermore, in RAW264.7 cells after coculture with calcitriol-treated Th17 cells inflammation-induced osteoclastogenesis was suppressed [303].

The above-mentioned studies analyzing the influence of vitamin D on Th17 cells have generally reported the inhibitory effect of vitamin D on the differentiation of these cells. Similarly, Chen et al. revealed that in vitro facilitated proliferation, migration and invasion of MCF-7 human breast cancer cells co-cultured with Th17 cells could be reversed by calcitriol. In these studies, calcitriol inhibited the differentiation of Th17 cells [304].

However, in our studies on 4T1 model, calcitriol and its analogs (PRI-2191 and PRI-2205) accelerated the metastatic potential of 4T1 tumor in young mice [144]. The mRNA screening showed increased expression of some genes associated with Th17 cells in the mononuclear splenocytes of young mice: IL-17a (*Il17a*), IL-17 receptor E (*Il17re*), *Il1r1*, *Il21*, RAR-related orphan receptor α (*Rora*), and RAR-related orphan receptor γ (*Rorc*) [282]. In parallel, we observed that in old OVX mice bearing 4T1 tumors, the same scheme of treatment leads to temporal decrease in lung metastases [151]. In the early stages after the 4T1 cells inoculation, calcitriol stimulated the expression of almost all genes tested related to Th17 differentiation in spleens of aged OVX mice. The effect of the tested compounds on the expression of Th17 cells associated genes in the advanced stage of the disease was not so significant, except for the *Il17re* gene. Interestingly, the expression of *Rorc* was diminished in calcitriol-treated aged OVX mice [305]. Analyses performed on unstimulated splenocytes showed some differences between young and old OVX mice. However, for detailed analysis of Th17 response, we stimulated CD4+ splenocytes harvested from young and aged OVX mice bearing 4T1 tumors, which were treated with calcitriol and its analog with TGF-β and IL-6 to induce Th17 cells ex vivo. The induced Th17 (iTh17) cells from young mice splenocytes treated with PRI-2191 expressed significantly higher levels of *Il17a* mRNA as compared to iTh17 splenocytes obtained from control tumor-bearing mice. Furthermore, PRI-2191 mice treatment resulted in higher expression of transcription factors *Rora* and *Rorc* and IL-17R (*Il17re*) as well as IL-21 (*Il21*) mRNA in iTh17 cells. Also VDR mRNA (*Vdr*) as well as OPN (*Spp1*) was significantly increased in iTh17 cells obtained from young mice after PRI-2191 treatment. On the other hand, aged OVX mice did not show any significant changes in the expression of the analyzed genes [305].

iTh17 cells from young, calcitriol analog-treated mice released higher levels of IL-17A than control, nontreated mice, while the opposite effect (decreased IL-17A secretion) in old OVX mice was observed [305]. IL-17 is a proinflammatory cytokine with proven pleiotropic effect. Its effect may vary depending on the stage of tumor development. The protumor effect of IL-17, supporting tumor angiogenesis, increases during the chronic phase of cancer and inflammation development, and overpowers anticancer effects, promoting the expansion of cytotoxic T lymphocytes and other immune cells fighting cancer [175,306]. In the 4T1 mouse model (young mice), in which calcitriol and its analog increased the IL-17A secretion of iTh17 cells [305], increased blood perfusion within the tumor was observed [144]. On the other hand, we didn’t observe any effect on tumor blood perfusion in aged OVX mice [151].

There is evidence of synergistic effect between E2 and vitamin D_3_. As it has been shown that vitamin D_3_ increases the synthesis of estrogens which are essential for the expression and function of VDR in the inflammation of the central nervous system [307]. Interestingly, our results according to studies on 4T1 mouse mammary gland cancer revealed an increase in plasma estrogen levels after calcitriol treatment only in young, tumor bearing mice [144], whereas both tested analogs (PRI-2191 and PRI-2205), and to a lesser extent, calcitriol decreased estrogen level in plasma of aged OVX mice bearing 4T1 tumors [305]. However, varying estrogen regulation in both groups of mice was not directly related to the expected impact on Th17 cells, as it has been previously reported that estrogen inhibits Th17 cell differentiation and IL-17 production inhibiting RORγT expression [196]. As we have shown, young mice treated with calcitriol analogs revealed increased plasma estrogen levels as well as increased *Rorc* (encoding RORγT) mRNA in iTh17 splenocytes. However, in old mice no such changes in expression were observed [305]. It is supposed that the hormonal status of the tumor-bearing organism did not have a direct effect on Th17 cells in our experimental conditions. However, after treatment with PRI-2191, iTh17 lymphocytes isolated from the spleens of young mice showed increased expression of *Vdr*, while an opposite trend was observed in aged OVX mice model. Therefore, it can be assumed that the hormonal status of the host organism in the 4T1 model influences the action of calcitriol and its analogs on *Vdr* expression in iTh17 cells, and that the expression of genes involved in Th17 cell formation by these compounds correlates with the level of *Vdr* expression [144,305]. In addition, after PRI-2191 treatment, an increased expression of the *Spp1* gene (encoding OPN) was observed in young 4T1 tumor-bearing mice [305]. According to the current knowledge, OPN is required by DCs for inducing Th17 cell differentiation [189] and IL-17 synthesis [190]. In acute coronary syndrome studies, OPN has been shown to correlate positively with inflammation, through a direct effect on IL-17-producing cells [191]. This direct effect of OPN on Th17 cell differentiation is the result of the interaction with its receptors [192]. Our studies revealed increased *Spp1* expression in iTh17 cells in young mice which correlated with the intensity of expression of the genes associated with the phenotype of these proinflammatory cells [305]. In addition, a significant increase in the OPN level after calctriol or its analogs treatment was observed in tumor tissue of young mice bearing 4T1 tumor in our previous studies [144], whereas in aged OVX mice its level in tumor tissue was significantly diminished [151]. Calcitriol directly stimulates the expression of OPN in various cells through VDR-responsive elements in *Spp1* gene [308,309]. Therefore, its increase in 4T1 tumor tissue [144], lymph nodes [282], or iTh17 cells from young mice may have contributed to the enhanced iTh17 cell differentiation and IL-17A production observed in young mice [305].

The actions of vitamin D analogs and metabolites are mediated through VDR. When occupied by a ligand, VDR heterodimerizes with the retinoid X receptor (RXR) and, together with coregulatory proteins, interacts with specific DNA sequences (vitamin D response elements) in the promoter regions of target genes, modulating their transcription [310]. We hypothesized that vitamin D could indirectly regulate Th17 cell differentiation through the impact of VDR on OPN (OPN is known to regulate the expression of IL-17 through its receptors [192] and OPN gene possesses VDR-responsive elements [309]), and this could affect the progression of the tumor (Figure 7).

## 6. Conclusions

Although many studies on vitamin D have been performed so far, it is unclear whether vitamin D (or its derivatives) reveals a beneficial or harmful effect in breast cancer treatment, and how useful vitamin D-containing diets and supplements really are. Therefore, it is necessary to investigate in-depth about the action of vitamin D, which, through its effect on the entire host organism, may indirectly influence tumor development and metastasis formation, even in the case where cancer cells are not themselves sensitive to vitamin D. Moreover, more attention should be paid to several receptors expressed by breast cancer cells (especially isoforms of both ER receptors) at different stages of tumor development. This should always be compared with the results for patients vitamin D and OPN blood status and the level of expression for enzymes metabolizing vitamin D in tumor tissue for all breast cancer cases, to make the knowledge most helpful in personalizing cancer patient treatment.

## Figures and Tables

**Figure 1 cancers-14-03649-f001:**
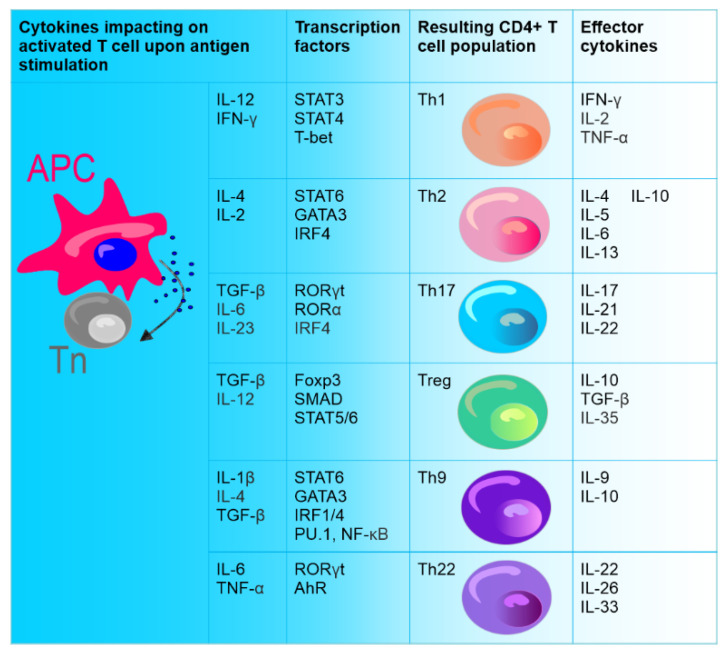
Differentiation of naïve CD4+ T cells (Tn). Naive CD4+ T cells (stem cell-like cells), under the influence of different cytokines secreted upon direct contact with antigen-presenting cell (APC), can differentiate into various types of effector cells: Th1, Th2, Th9, Th17, Th22, and Treg. CD4+ T cell subsets are defined by the production of specific cytokines and the expression of specific transcription factors.

**Figure 2 cancers-14-03649-f002:**
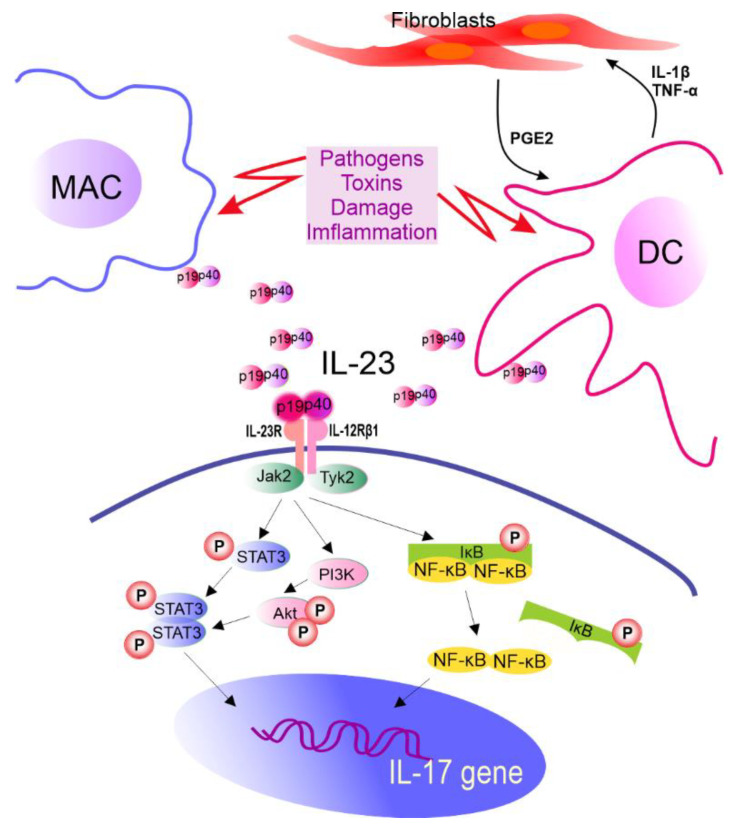
The role of IL-23 in IL-17 gene expression. IL-23R pairs with IL-12Rβ1 forming IL-23R complex required for IL-23 signaling. This receptor is constitutively associated with Janus kinase 2 (Jak2) and Tyrosine kinase 2 (Tyk2) which are activated after ligand biding, leading to STAT3 phosphorylation (P). Other molecules in IL-23 signaling cascade are also identified. MAC—macrophages, DC—dendritic cells, IL—interleukin, PGE2—prostaglandin E2, TNF-α—tumor necrosis factor α, STAT3—signal transducer and activator of transcription 3, PI3K—phosphoinositide 3-kinase, Akt—serine/threonine-protein kinase, NF-κB—nuclear factor kappa-light-chain-enhancer of activated B cells, IκB—NF-κB inhibitor, and p19 and p40—subunits of IL-23.

**Figure 3 cancers-14-03649-f003:**
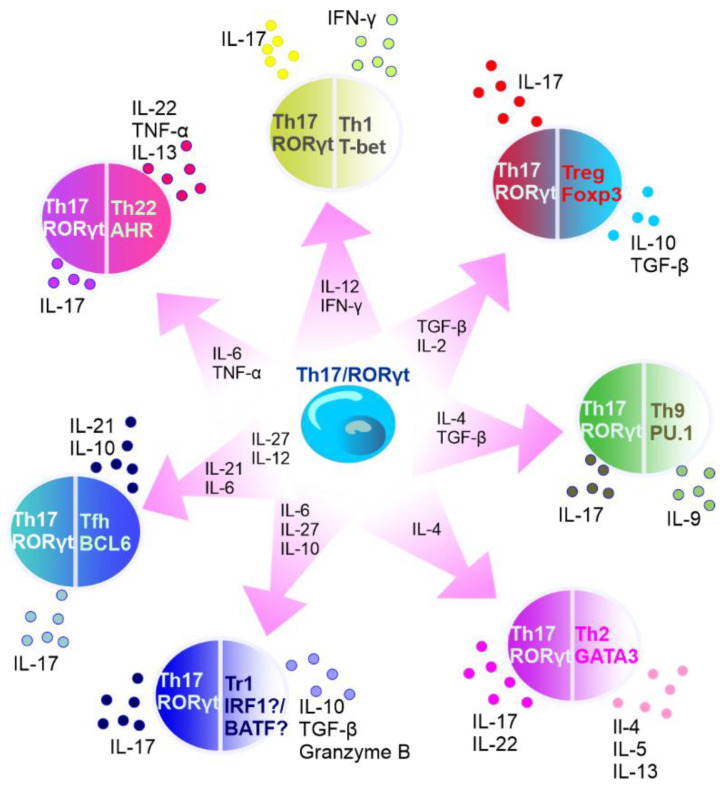
Summary of Th17 cell plasticity. In the circles, the names of T cells are presented with main transcription factors listed below the names. Outside the circles, the cytokines secreted by the specific cells are listed. Transcription factors: AhR—aryl hydrocarbon receptor, RORγt—retinoid-acid-receptor-related orphan nuclear receptor γ, IRF1—interferon regulatory factor 1, PU.1—transcription factor encoded by SPI1 gene, BATF—basic leucine zipper transcriptional factor ATF-like, BCL6—B-cell lymphoma 6, T-bet—member of T-box family of transcription factors, GATA3—GATA binding protein 3, and FOXP3—forkhead box P3. Cytokines: IL—interleukin, TNF-α—tumor necrosis factor α, IFN-γ—interferon γ, and TGF-β—transforming growth factor β. Cells: Tr1—type 1 regulatory T cells, Tfh—T follicular helper cells, Th—T helper cells, and Treg—T regulatory cells.

**Figure 4 cancers-14-03649-f004:**
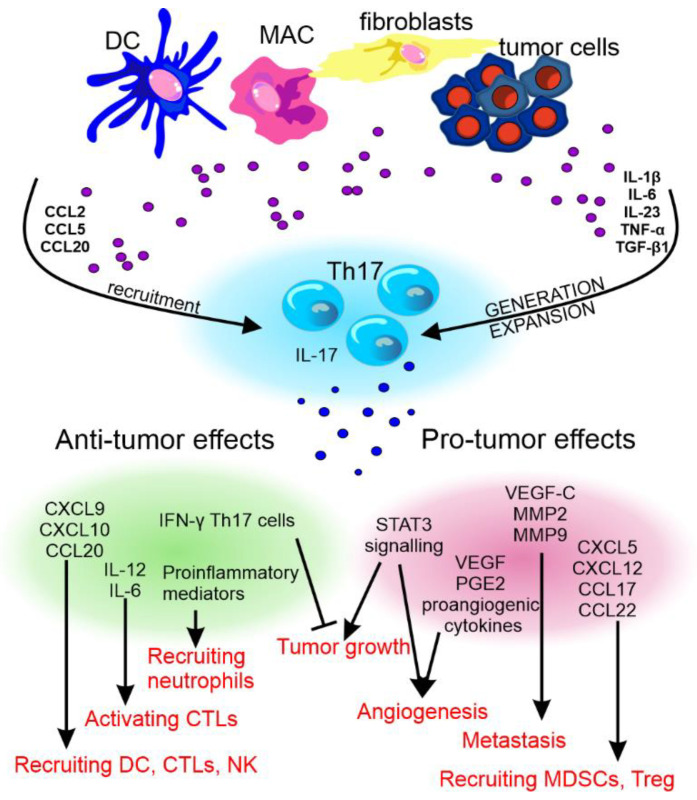
The anti- and pro-tumor effects of Th17 cells. The recruitment and differentiation of Th17 cells in the tumor environment is influenced by factors produced by dendritic cells (DCs), macrophages (MAC), fibroblasts, and cancer cells. Th17 cells differentiated in this way may show various effects on tumor development. CCL—C-C motif chemokine ligand, TNF-α—tumor necrosis factor α, TGF-β—transforming growth factor β, CXCL—C-X-C motif chemokine ligand, MMP—matrix metalloproteinase, STAT3—signal transducer and activator of transcription 3, VEGF—vascular endothelial growth factor, PGE2—prostaglandin E2, IFN-γ—interferon γ, IL—interleukin, CTLs—cytotoxic T lymphocytes, and NK—natural killer cells.

**Figure 5 cancers-14-03649-f005:**
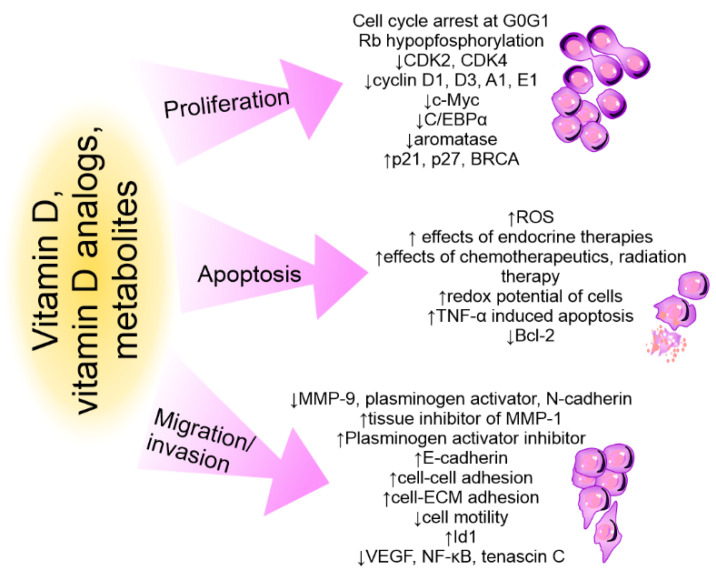
Summary of the mechanisms of vitamin D direct action on breast cancer cells. Rb—retinoblastoma, CDK—cyclin-dependent kinase, c-Myc—c-myelocytomatosis, C/EBPα—CCAAT-enhancer-binding protein α, p21 and p27—cyclin-dependent kinase inhibitors, BRCA—breast cancer susceptibility gene, ROS—reactive oxygen species, TNF-α—tumor necrosis factor α, Bcl-2—B-cell leukemia/lymphoma 2, MMP—matrix metalloproteinase, ECM—extracellular matrix, Id1—inhibitor of DNA binding 1, VEGF—vascular endothelial growth factor, and NFκB—nuclear factor kappa-light-chain-enhancer of activated B cells.

**Figure 6 cancers-14-03649-f006:**
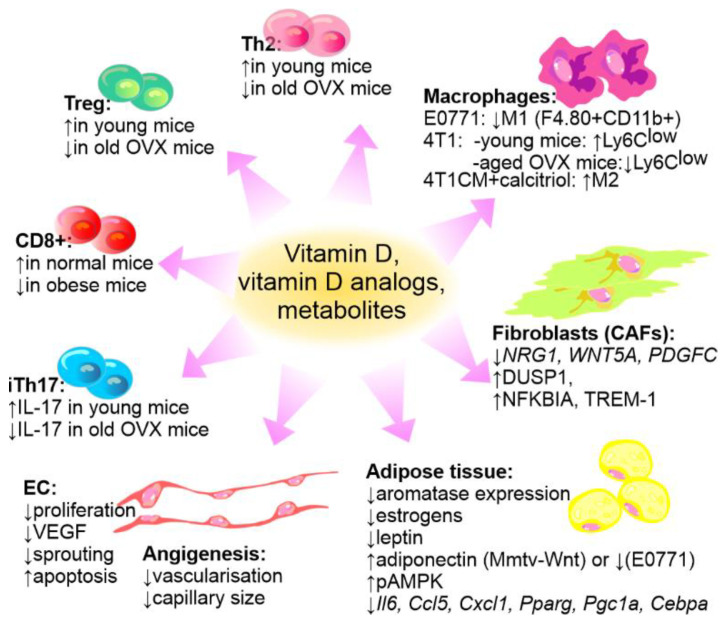
Summary of the effects of vitamin D on various cells in tumor microenvironment and on immune system in breast cancer. CAFs—cancer-associated fibroblasts, iTh17—induced Th17, EC—endothelial cell, Treg—regulatory T lymphocytes, CD8^+^—CD8^+^ T lymphocytes, Ly6C—lymphocyte antigen 6 complex, *NRG1*—neuregulin 1, *WNT5A*—Wnt family member 5A, *PDGFC*—platelet-derived growth factor C, *DUSP1*—dual specificity phosphatase 1, *NFKBIA*—NFκB inhibitor α, *TREM-1*—triggering receptor expressed on myeloid cells 1, *Il6*—interleukin 6, *Ccl5*—C-C motif chemokine ligand 5 (RANTES—Regulated on Activation, Normal T-cell Expressed and Secreted), *Cxcl1*—C-X-C motif chemokine ligand 1, *Pparg*—peroxisome proliferator-activated receptor γ, *Pgc1a*—PPARγ coactivator 1α, *Cebpa*—CCAAT enhancer-binding protein α, VEGF—vascular endothelial growth factor, OVX—ovariectomized, AMPK—5′-AMP-activated protein kinase, E0771, 4T1, Mmtv-Wnt—mouse mammary gland cancer cell lines.

**Figure 7 cancers-14-03649-f007:**
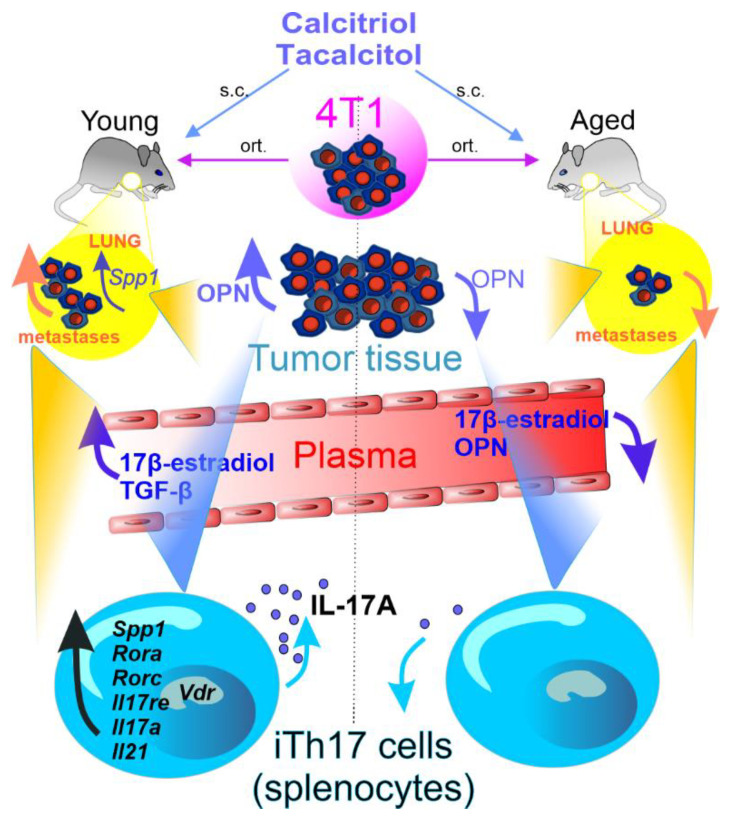
The effect of calcitriol or its analogs on Th17 cells in young and aged mammary gland tumor-bearing mice. Young (~8 weeks old) or aged ovariectomized (~50 weeks old) BALB/c mice were transplanted orthotopically (ort.) with 4T1 mouse mammary gland cancer cells, and then injected s.c. with calcitriol or tacalcitol. Increased metastatic potential was observed in young mice treated with calcitriol or tacalcitol, whereas in aged mice decreased metastasis count was noticed. In young mice, lung tissue showed increased osteopontin gene (*Spp1*) expression with increased osteopontin (OPN) protein level and plasma level of 17β-estradiol and TGF-β, which may contribute to increased Th17 cell differentiation and increased metastatic spread of 4T1 tumors. In aged mice, OPN level in tumor tissue, as well as plasma level of OPN and 17β-estradiol, was decreased with lower IL-17A production by induced Th17 cells (iTh17), leading to decreased metastatic potential of 4T1 cells.

**Table 1 cancers-14-03649-t001:** Examples of clinical/epidemiological data regarding the impact of vitamin D on breast cancer incidence/progression/metastasis.

Factor Studied	Type of Studies	Observed Effects	References
Sunlight exposition	Case–control	Sunlight measures are not associated with breast cancer risk but may depend on timing of exposure and genetic background	[34]
	Prospective	UV radiation has no association with breast cancer risk	[35]
	Systematic review and meta-analysis	Exposure to sun for longer than an hour a day during the summer could decrease the risk of breast cancer	[36]
	Cancer database analysis	Breast cancer is diagnosed more often in spring and fall; breast cancer seasonality is latitude-dependent	[37]
	Retrospective analysis	Sunlight exposure may be associated with more prevalent TNBC	[38]
	Population-based	Spending more daylight hours outdoors in a year was associated with lower risk of ER+, ER−, and TNBC	[39]
	Population-based case–control	Factors suggestive of increased cutaneous production of vitamin D are associated with reduced breast cancer risk	[40]
	Death certificate based case–control	Breast cancer was negatively associated with residential and occupational sunlight	[41]
Vitamin D dietary intake	Population-based	In the pooled analysis, dietary vitamin D and calcium were not associated with risk of breast cancer subtypes. Vitamin D: possible inverse associations between intake of ≤800 IU/d (compared with nonuse) and risk of several subtypes, with strongest effects observed for TNBC; no association was found for >800 IU/d	[39]
	Prospective	Higher intakes of calcium and vitamin D may be associated with lower risk of premenopausal breast cancer, but not among postmenopausal women	[42]
	Nutrition cohort	Women with the highest intake of dietary calcium (>1250 mg/d) were at a lower risk of breast cancer than ≤500 mg/d; but neither use of supplemental calcium nor vitamin D intake was associated with breast cancer risk	[43]
	Double-blind, placebo-controlled clinical trial and cohort study	Only some evidence for a reduction in breast cancer risk and total invasive cancer risk among calcium and vitamin D users	[44]
	Case–control	High calcium, phosphorus, and vitamin D nutrient intake pattern was associated with a significant decrease in breast cancer risk	[45]
	Case–control	Supplementation with vitamin D, fatty acids EPA, and DHA was inversely associated with breast cancer	[46]
	Case–control	The risk of breast cancer was lower by 67% if the serum level of 25(OH)D was ≥24.6 ng/mL and lower by 68% if the serum level of calcium was ≥9.6 mg/dL; higher (than-normal) calcium serum level, considered separately, and a slightly lower-than- normal vitamin D serum level may protect against breast cancer among postmenopausal women, independent of dietary patterns and supplements	[47]
25(OH)D plasma level	Observational	Low 25(OH)D serum levels, alone and in combination with BsmI VDR genotype, may increase the risk of breast cancer in a UK Caucasian population	[48]
	Cross-sectional analytical study	25(OH)D deficiency is widespread among breast cancer patients	[49]
	Nested case–control	No overall association was found between plasma 25(OH)D and breast cancer risk; women with high, compared with low, plasma 25(OH)D levels in the summer have a reduced breast cancer risk, and plasma 25(OH)D may be inversely associated with risk of tumors expressing high levels of VDR	[50]
	Case–control	Low serum 25(OH)D levels, high tissue VDR levels, and high ERα gene expression were associated with increased risk of breast cancer	[51]
	Unmatched case–control	Severe vitamin D deficiency (<25 nmol/L) was significantly higher in chemotherapy-naïve (41.1%) than in TAM-treated (11.2%) patients; vitamin D deficiency was not significantly associated with tumor characteristics or *VDR* genotype	[52]
	Nested case–control	No association was found between plasma level of free 25(OH)D and overall risk of breast cancer; no association was found for plasma vitamin D binding protein (DBP) as well; neither the total nor the calculated free 25(OH)D and breast cancer association substantially varied by plasma DBP levels	[53]
	Cross-sectional	Significant correlation was observed between low vitamin D levels and advanced stage of breast cancer, particularly in postmenopausal patients	[22]
	Prospective	Maintaining an optimal 25(OH)D status at diagnosis and during the 1-year follow-up period is important for improving breast cancer patient survival	[21]
	Prospective	Serum levels of 25(OH)D were significantly higher in patients with early-stage breast cancer than in those with locally advanced or metastatic disease	[23]
	Case–control	Women with low levels of 25(OH)D, as compared to women in the middle tertile, had a high risk of breast cancer with an unfavorable prognosis	[26]
	Case–cohort	In women with elevated risk of breast cancer, high serum 25(OH)D levels and regular vitamin D supplement use were associated with lower rates of postmenopausal breast cancer over 5-year follow-up	[54]
	Cross-sectional	Low and decreased level of vitamin D might correlate with progression and metastasis of breast cancer	[27]
	Retrospective cohort study	25(OH)D serum level < 16 ng/mL was associated with poor survival in breast cancer patients, regardless of age, lymph node status, stage, or breast cancer subtype	[55]
	Single-center prospective cohort study	Low 25(OH)D status is associated with better breast cancer survival; high 25(OH)D levels (>110 nmol/L) are associated with poorer breast cancer survival	[33]
	Meta-analysis	Lower 25(OH)D concentrations were not significantly associated with increased incidence of most cancers assessed; increased risk of breast cancer and decreased risk of lymphoma were noted with higher 25(OH)D concentrations	[32]
	Systematic review and meta-analysis	Association of low levels of vitamin D with increased risk of recurrence and death was noted in early-stage breast cancer patients	[56]
	Meta-analysis	A protective relationship was observed between circulating vitamin D (measured as 25(OH)D) and breast cancer development in premenopausal women	[31]
	Genome-wide association studies	No evidence was found supporting the association between 25(OH)D and risk of breast cancer	[57]
	Retrospective	Vitamin D deficiency was associated with inability to achieve complete pathological response following neoadjuvant chemotherapy	[58]
	Pooled cohort	Higher 25(OH)D concentrations, ≥60 ng/mL, were associated with a dose–response decrease in breast cancer risk.	[59]
1,25(OH)_2_D_3_ plasma level	Prospective	breast cancer patients have lower 1,25(OH)_2_D_3_ (40 ± 21) levels than healthy women (53 ± 23 pg/mL)	[60]
VDR tumor tissue expression	Observational	VDR expression inversely related to aggressive tumor characteristics (large tumor size, hormonal receptor negativity, and triple-negative subtype); VDR expression did not influence any patient survival outcomes	[28]
	Retrospective	VDR expression was negatively associated with tumor size and lymph node involvement. High VDR expression speaks for better patient outcome	[61]
	Prospective cohort study	High VDR expression in invasive breast tumors was associated with favorable prognostic factors and lower risk of breast cancer death	[29]
	Case–control	VDR was upregulated in breast cancer tissues especially in hormone-negative breast cancer	[51]
	State-of-the-science review	The role of VDRs in cancer etiology is still equivocal	[62]
VDR CTCs expression	Observational	VDR+ CTCs were detected in 46% of CTC+ patients	[30]
CYP27B1/CYP24A1 tumor tissue expression	Observational	mRNA expression of CYP27B1 was downregulated in tumor tissues, compared with normal tissues; the mRNA expression of CYP24A1 was significantly upregulated in the tumor tissues	[63]
	Observational	Expression of CYP24A1 mRNA was reduced by about 58% in breast cancer tissues	[64]
	Observational	CYP27B1 expression was lower in invasive carcinomas (44.6%) than in benign lesions (55.8%). In contrast, CYP24A1 expression was inreased in carcinomas (56.0% in in situ and 53.7% in invasive carcinomas) compared to benign lesions (19.0%)	[18]
	Prospective	No correlation was observed between 24-hydroxylase, 1α-hydroxylase, and VDR expression in tumor tissue	[60]
Vitamin D-binding protein	Meta-analysis	Borderline decrease in cancer risk was noted for subjects with high levels of DBP	[65]
Vitamin D-related gene polymorphism	Case–control	Two *CYP24A1* polymorphisms (rs34043203, rs2762934) were associated with increased breast cancer risk; one with reduced breast cancer risk (rs1570669); minor alleles for the *VDR Bsm1* polymorphism (rs1544410) but not *Fok1* (rs2228570) were inversely associated with breast cancer risk	[34]
	Population-based, case–control	None of the analyzed polymorphisms (*Fok*I and *Taq*I) were associated with overall risk for postmenopausal breast cancer; *Taq*I polymorphism–a significantly increased risk for ER+ tumors but not for ER–tumors; haplotype FtCA was associated with a significantly greater breast cancer risk as compared with the most frequent haplotype FTCG	[66]
	Large population-based case–control	Increased risk for breast cancer was found in postmenopausal Caucasian women with the *Bsm*I *bb* genotype	[67]
	Case–control	The VDR *Tru9*I “uu” genotype may increase the risk of premenopausal breast cancer	[68]
	Case–control	*VDR Fok*I *ff* (rs2228570) polymorphism was significantly associated with an increased risk of breast cancer	[69]
	Unmatched case–control	*VDR* common variant alleles rs7975232 (*Apa*I), rs2228570 (*Fok*I), and rs731236 (*Taq*I) were analyzed; rs2228570 *GG* genotype was associated with increased risk of breast cancer	[52]
	Case–control	GG genotype of *Cdx2-VDR* gene polymorphism may increase the risk of developing breast cancer in young female patients in South Pakistan	[70]
	Observational	VDR gene polymorphisms (*Bsm*I and *Apa*I) may contribute to breast cancer risk among postmenopausal women	[71]
	Case–control	Significantly increased risk of breast cancer was associated with *BsmI bb* or even *Bb* genotype. Significant association between *Fok*I genotypes and breast cancer risk was not observed	[72]
	Comparative	Early-onset patients revealed an association between rs10735810 and increased breast cancer risk; rs1544410, rs731236, and rs4516035 showed no association with disease	[73]
	Case–control	*BsmI* polymorphism in VDR gene may be associated with an increased breast cancer risk in Pakistani women negative for BRCA1/2 germline mutations	[74]
	Multicenter, prospective	Odds ratio for the rs2228570 (*Fok*I) *ff* versus *FF* genotype in the overall population was statistically significantly elevated; no association; *BB* genotype was associated with a significantly lower risk of advanced breast cancer	[75]
	Case–control correlative	Vitamin D levels were not significantly associated with development of AI-induced arthralgia (AIA); patients with *Fok*I *VDR* genotype were less likely to develop AIA	[76]
	Systematic review and network meta-analysis	Recessive polymorphism model with the rs2228570 (*Fok*I) polymorphism is the best predictor of breast cancer in Caucasian patients; homozygote model with the *CDX2* polymorphism is the best predictor of breast cancer in African-American patients	[77]
	State-of-the-science review	VDR polymorphisms may affect the risk and mortality of breast cancer, according tomenopausal status, vitamin D level, and breast cancer risk and mortality	[62]
	Systematic review of the literature	VDR polymorphisms Fok1, Bsm1, Taq1, Apa1, and Cdx2 were analyzed, but no; conflicting data were obtained	[78]

**Table 3 cancers-14-03649-t003:** Summary of animal experiments in breast cancer models with the use of vitamin D, calcitriol, or its analogs, alone or combined with anticancer therapies.

Animal Model	Vitamin D Metabolite/Analog Used	Schedule of Treatment	Monotherapy/Combined Treatment/Other	Effect Observed	References
4T1 mouse mammary gland cells	Vitamin D_3_ (VD)	Day 17 after tumor inoculation, 5 μg/kg of VD/d, 7 days treatment	-	Increase in tumor growth	[142]
4T1, 67NR, E0771 mouse mammary gland cancers	Calcitriol (Cal) 1 µg/kg three times a week p.o. or 5000 IU of vitamin D_3_/kg of diet	Cal: 7 days after implantation till day 21; Diet: 6 weeks before transplantation and continued	-	4T1: increased metastasis; 67NR: no effect; E0771: decreased tumor growth	[145]
4T1 mouse mammary gland cells	Calcitriol, PRI-2191, and PRI-2205 0.5, 1, and 10 µg/kg, respectively, three times a week s.c.	7 days after implantation till day 33	Postmenopausal OVX model	Transiently decreased metastasis	[151]
4T1 mouse mammary gland cells	Calcitriol, PRI-2191, and PRI-2205 0.5, 1, and 10 µg/kg, respectively, three times a week s.c.	7 days after implantation till day 33	-	Increased metastasis	[144]
E0771 mouse mammary gland cancer	Cholecalciferol gavage 40 IU/d/mouse	7 days after cell injection for 2 weeks	Normal mice Obese mice	Antitumor, antimetastatic effect Stimulation of tumor growth and metastasis	[152]
LM3 mouse breast adenocarcinoma	EM1 20 and 50 µg/kg	Started when tumors were 50–70 mm^3^, 7 injections for 2 weeks	-	Tumor growth—no effect; decrease in lung metastasis	[153]
ER–breast cancer/ER+ breast cancer	24R,25(OH)_2_D_3_	HCC38, 2 weeks after inoculation, 24R,25(OH)_2_D_3_ 100 ng/d, 3 times a week, 10-week treatment MCF-7: 25 ng or 100 ng per injection of 24R,25(OH)_2_D_3_ from week 5	-	Protumorigenic when breast cancer was of Erα-66– and Erα-36 ± status: the response of breast cancer cells to 24R-24,25(OH)_2_D_3_. Enhanced or decreased epithelial-to-mesenchymal transition	[154]
4T1 mouse mammary gland cells	Calcitriol 0.3 μg/kg b.w. once every other day i.p.	From the day before tumor cells were injected	-	Antimetastatic	[155]
16/c mouse mammary adenocarcinoma	Calcitriol or PRI-2191 10 μg/kg for 5 consecutive days s.c.	From day 5 after tumor inoculation	Cisplatin: 3 mg/kg i.p. at day 6 after tumor cell inoculation; clodronate: days 5 and 8, i.p., 1.5 mg/mouse/d	54% tumor growth inhibition by combined treatment with PRI-2191; 41% inhibition by treatment with PRI-2191 alone; no effect of calcitriol	[156]
16/c mouse mammary adenocarcinoma	PRI-1906 or PRI-2191 1 μg/kg/d for 9 consecutive days s.c.	Started from day 1 or 5 after tumor cell inoculation	Cyclophosphamide (CY) i.p. 100 mg/kg on day 4 after tumor cell inoculation	Potentiation of CY antitumor effect by both analogs. PRI-1906 alone—no effect; PRI-2191—25% of inhibition	[157]
MDA-MB-231-luc	Calcitriol 1 µg/kg	3 days prior to photodynamic therapy (PT)	5-Aminolevulinate-based PT	Increased PT effect *	[158]
MMTV-Wnt1 mouse mammary gland cells	Calcitriol 25 ng/mouse or 5300 IU of vitamin D_3_/kg of diet	10 weeks after standard (STD) and high-fat (HFD) diet	STD and HFD	Decreased tumor volume	[159]
MMTV-Wnt1 mouse mammary gland cells	Calcitriol 50 ng/mouse three times a week or 5000 IU of vitamin D_3_/kg of diet	8 weeks prior to tumor inoculation and continued	-	Decreased tumor volume	[134]
MCF-7 human breast cancer	Calcitriol 50 ng/mouse three times a week i.p. or 5000 IU of vitamin D_3_/kg of diet	After 6 weeks of tumor growth continued for 4 weeks	-	Decreased tumor growth	[160]
MCF-7 human breast cancer	Calcitriol 0.025, 0.05, or 0.1 μg/mouse, three times a week i.p. or 5000 IU of vitamin D_3_/kg of diet	After 6 weeks of tumor growth continued for 4 weeks	Pre- and postmenopausal (OVX) models	Decreased tumor growth (~60%)	[161]
MCF-7 human breast cancer	PRI-2191 and PRI-2205 1.0 μg/kg/d or 10.0 μg/kg/d, respectively, three times a week s.c.	From day 39 after tumor cell inoculation up to day 67	-	Decreased tumor growth by PRI-2205	[136]
MCF-7 (VEGF-transfected) with MDA-435S human breast cancer cells	Calcitriol 12.5 pmol/d s.c. (micro-osmotic pumps)/28 days; next weekly s.c. dose of 12.5 pmol/d 7 times	6 days before cell implantation and during 8 weeks	-	Decreased tumor angiogenesis	[162]
MCF-7 human breast cancer	Calcitriol and analogs: TX 522 and TX 527 5, 80, and 25 μg/kg, respectively, every other day i.p.	Started 4 days after tumor transplantation	-	Cal: no effect; both analogs: decrease in tumor volume and mitotic figures	[163]
MCF-7 human breast cancer	EB1089 45 pmol/d (osmotic pumps)	Started with 150–200 mm^3^ tumors for 8 days	Irradiation after end of EB1089 (2 × 5 Gy)	Delayed tumor growth and decreased tumor volume (50%) in combined treatment	[164]
MCF-7 human breast cancer	PRI-2191 1.0 µg/kg b.w./d, PRI-2205 10.0 µg/kg b.w./d	Started 5 days after tumor cell inoculation	Anastrozole: 5 days/week, 200 µg/mouse in each injection	Significantly decreased MCF-7 tumor volume after single or combined treatment	[147]
T47D or TDC human breast cancer	Calcitriol 0.03 μg/kg, 2 times a week i.p.	Started when tumors were palpable during 4 weeks	-	Decreased tumor growth, no effect on angiogenesis	[165]
MCF10DCIS.com human DCIS model	BXL0124 0.1 μg/kg, 6 times a week i.p. or 0.03 or 0.1 μg/kg, 6 times a week p.o. or i.p.	Next day after tumor inoculation for 4 weeks or for 5 weeks	-	Inhibition of DCIS to IDC progression; 43% reduction in tumor volume	[166,167]
SUM149 human inflammatory breast cancer cell line	Quantum dots with calcitriol 40 nM i.v.	Started with 80 mm^3^ tumors	Quantum dots with calcitriol conjugated with anti MUC-1 Ab	Enhanced concentration of quantum dots in tumor and lungs	[168]
Transgenic model of hormone-induced mammary cancer (LH-overexpression)	EB1089 0.027 μg per animal s.c.	From 3 to 5 weeks of age	-	Decreased growth to regression	[169]
Freshly collected breast cancer samples xenografted into animals	Calcitriol 0.06 μg intratumoral	6 weeks after transplantation, weekly (6–11)	-	No effect	[170]
N-methyl-N-nitrosourea (NMU)-induced mammary tumor (rats) and MCF10DCIS.com	Gemini 0072 and Gemini 0097 0.1, 0.3, or 0.03 μg/kg, 5 days a week i.p.	1 week after NMU; day 4 after MCF10DCIS.com cell implantation		60% inhibition of NMU-induced mammary tumor and suppression of MCF10DCIS.com	[171]
NMU- and DMBA (7,12-dimethylbenz(a)anthracene)-induced mammary carcinogenesis	1α-Hydroxy-24-ethylcholecalciferol 25, 40, 50 μg/kg of diet	2 weeks before carcinogen	-	Chemopreventive effects	[172]

* Calcitriol alone stimulates differentiation and proliferation in MDA-MB-231-luc tumors.

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
