# Peer review of "Vitamin D, Th17 Lymphocytes, and Breast Cancer"

_cancers, 2022, doi:10.3390/cancers14153649_

Round 1

Reviewer 1 Report

Dear Authors,

This is an interesting review.

Here are my observations/suggestions/comments:

1.    Abstract – Conclusion – Despite the fact that VD effects depends on many factors, you should mention a few clear ideas (take-home messages)

2.    Introduction - Line 33- These are not the causes, but some risk factors depending on different studies

3.    “A study analyzed the serum level of 25-hydroxyvitamin D (25(OH)D; a metabolite used to assess the vitamin D status in the body) in pre- and postmenopausal women, as well as genetic predisposition, such as VDR gene polymorphism and expression of vitamin D-metabolizing enzymes at various stages of cancer development [15].”  There is not conclusion of this study.

4.    Also, 25-hydroxyvitamin D is the most important parameter in order to assess the vitamin D status, not any “metabolite”. This is generally applicable, regardless age, sex, menopausal status.

5.    Line 103-125. Please re-structure the paragraph. You switch form historical irrelevant data to mentioning studies that actually are insignificant among the large panel of VD studies. Then you turn to some classical physiological data concerning skin synthesis of VD.

6.    Chapters 2.1-2.2.-2.3. You should not mix the data: there are studies on 25-hydroxyvitamin D levels and cancer risk and another large category of studies involves VDR which are mostly not correlated with the current assessments on VD metabolism.

I suggest you clearly separate these ideas into distinct micro-chapters:

1. The level of 25-hydroxyvitamin D and cancer correlations; including the subgroups with VD deficiency (here, there are 2 aspects: transversal studies and longitudinal studies, because the statistical significant is very different);

2. The studies on VDR;

3. The studies on intervention – meaning which is the potential effect on VD supplementation (in patients with VD deficient status or not) on cancer progression. Here you should mention, also, the studies on different daily VD intake.

4. Pre-clinical data, murine experiments, signal transduction data, etc.

I suggest you re-structure these chapters, including cited studies which associate different levels of statistical significance. Also, Table 1 should be reconsidered in addition to these micro-chapters

7.    Table 1. Sun exposure and risk of breast cancer underling VD status is a heterogeneous subject and the presentation seems unclear. The fact that someone has sun exposure does not mean that VD deficiency is not present since UB protection might be use. On the other hand, VD supplements correct a VD deficiency in persons without sunlight exposure.  

8.    Conclusion – You should not cite any reference at conclusion

9.    I suggest you finish at discussions or even conclusion with your own ideas, concepts based on the large amount of data you analyzed, including limits of current knowledge, controversies, new ideas to develop, and new directions of studies in order to provide clear data.

Best regards,

Thank you,

Author Response

Dear Reviewer,

Thank you very much for all valuable comments, your time spent on our paper and for the work put into this review. In the main text of the manuscript you will find all the corrections I’ve done. I hope that the changes I’ve made will increase the value of our article. Below, in the attached file, you will find answers/explanations to your question/comments written in red.

Reviewer 2 Report

It's a good compile of information and studies with minor changes to perform in the style. Also, a few considerations need to be pointed out,  this review has many studies on vitamin D and its relationship directly or indirectly influence tumor development in breast cancer. Still, many of these studies performed so far are unclear that need to be addressed in the future. Still, it's a good compilation of information for other investigators to preview the potential effects of vitamin D in the progression of breast cancer.

Author Response

Dear Reviewer,

Thank you very much for your time spent on our paper and for the work put into this review and your comments. In the main text you will find all the corrections I’ve done. I hope that the changes I’ve made will increase the value of our article.

Thank you for all comments. I agree, there is still much work to be done in the field of vitamin D in breast cancer.

                                                                                   Kind regards,

                                                                                   Beata Filip-Psurska
